# LEARNING TO SEARCH FROM DEMONSTRATION SEQUENCES

**Dixant Mittal**[1,2]    **Liwei Kang**[1]    **Wee Sun Lee**[1]
[1]National University of Singapore
[2] Moovita
`{dixant,kang,leews}@comp.nus.edu.sg`

## ABSTRACT

Search and planning are essential for solving many real-world problems. However, in numerous learning scenarios, only action-observation sequences, such as demonstrations or instruction sequences, are available for learning. Relying solely on supervised learning with these sequences can lead to sub-optimal performance due to the vast, unseen search space encountered during training. In this paper, we introduce Differentiable Tree Search Network (D-TSN), a novel neural network architecture that learns to construct search trees from just sequences of demonstrations by performing gradient descent on a best-first search tree construction algorithm. D-TSN enables the joint learning of submodules, including an encoder, value function, and world model, which are essential for planning. To construct the search tree, we employ a stochastic tree expansion policy and formulate it as another decision-making task. Then, we optimize the tree expansion policy via REINFORCE with an effective variance reduction technique for the gradient computation. D-TSN can be applied to problems with a known world model or to scenarios where it needs to jointly learn a world model with a latent state space. We study problems from these two scenarios, including Game of 24, 2D grid navigation, and Procgen games, to understand when D-TSN is more helpful. Through our experiments, we show that D-TSN is effective, especially when the world model with a latent state space is jointly learned. The code is available at `https://github.com/dixantmittal/differentiable-tree-search-network`.

## 1 INTRODUCTION

Search and planning are critical components in a wide range of complex tasks. It has long been important in areas such as robotics and games (Mohanan & Salgoankar, 2018; Duarte et al., 2020), and has recently gained prominence in the area of large language models as inference-time solutions for improving reasoning and agentic abilities (Yao et al., 2023; Hao et al., 2023; Zhou et al., 2024). In many of these domains, step-by-step demonstrations are available in the form of recorded expert demonstrations, medical treatment records, and instruction/solution manuals. However, constructing a search tree would require data that is not directly available from the demonstration sequences – this can be impractical or very expensive to obtain. Hence, it would be desirable to learn to search from only demonstration sequences.

Search and planning can be effectively executed if a simulator or a world model is available. When such a world model is not available, we may learn the world model and then use it for these purposes (Kaiser et al., 2020; Hafner et al., 2020; Ha & Schmidhuber, 2018). However, when separately learned from demonstration sequences, such world models often suffer from compounding errors, and cannot cover the complete state space. Using such world models for search can be unreliable, particularly if the search expands into parts of the space that are not seen during training (Asadi et al., 2018).

One way to increase the reliability of the world model is to perform end-to-end learning (Farquhar et al., 2018), where the world model is trained with the search algorithm. One of the advantages of end-to-end learning is that the search algorithm will learn to avoid going into states where the world model is inaccurate since this would result in lower performance. We begin by showing that, when the search tree has a fixed structure, the loss function for training is continuous with respect to the

model parameters, justifying the use of a fixed-structure search tree with gradient-based learning methods. However, if the search tree structure can change during training, the loss function may no longer be continuous. This is because a small change in parameters could result in the construction of a different search tree and bring a large change in the corresponding loss function. To alleviate this issue, we propose employing a stochastic tree expansion policy and optimizing the expected loss, ensuring the continuity of loss function in the parameter space.

We further propose formulating the search tree expansion as another decision-making task with the goal of minimizing the prediction error progressively during the tree expansion and refining the tree expansion policy via REINFORCE (Williams, 1992). The use of the REINFORCE algorithm introduces another challenge; REINFORCE usually has high variance in its gradient estimates. To handle that, we propose the use of a baseline for variance reduction, adopting the telescoping sum trick used in Guez et al. (2018) for constructing the baseline.

Employing these techniques allows us to train a novel neural network, Differentiable Tree Search Network (D-TSN), that embeds the structure of a best-first search algorithm into its architecture. D-TSN can use a simulator or world model, if available, to learn to perform search. We study this scenario by fine-tuning a large language model to play the Game of 24 (Yao et al., 2023). D-TSN can also be used in more difficult scenarios where the world model is not available and only sequences of demonstrations are provided. We study two problems in this scenario, a small-scale 2D grid navigation problem and Procgen games (Cobbe et al., 2020) which are substantially more complex. Through our experiments, we show that end-to-end learning for doing search and planning from only sequences of demonstrations is possible and useful, especially when the world model needs to be jointly learned.

## 2 RELATED WORKS

Search and planning are crucial in various domains. For example, in language model reasoning, recent works integrated search algorithms to enhance the reasoning capabilities of language models. These techniques employ in-context queries (Yao et al., 2023; Xie et al., 2023; Hao et al., 2023; Zhou et al., 2024) or additional value heads (Liu et al., 2023) to guide the search process during inference, enhancing model's reasoning ability by looking at different possible solutions. In gaming, notable successes (Silver et al., 2016; Schrittwieser et al., 2020; Nasiriany et al., 2019) have combined deep neural networks with search algorithms to achieve superhuman performance.

One line of recent works has focused on embedding search inductive bias into network architectures. Neural Admissible Relaxation (NEAR) (Shah et al., 2020) develops approximately admissible heuristics for the A* algorithm. Neural A* Search (Yonetani et al., 2021) integrates the A* algorithm into network structures, learning a cost function from a gridworld map. Similarly, MCTSnets (Guez et al., 2018) incorporate the framework of Monte Carlo Tree Search (MCTS) into network architectures, guiding the search using parameterized memory embeddings stored in a tree structure. However, these methods rely on a known world model for planning, which limits their application when the world model is unknown. Another notable work, TreeQN (Farquhar et al., 2018), incorporates search inductive bias into the network by fully expanding a search tree to a fixed depth while jointly learning a world model, which allows it to tackle problems where the world model is unknown. However, the full expansion mechanism of TreeQN leads to a shallow search tree, which limits its application in complex problems with longer trajectories.

Our setting of learning from sequences of demonstrations is closely related to Offline Reinforcement Learning (Prudencio et al., 2024; Levine et al., 2020), where an agent learns its policy solely from a fixed dataset of experiences without further interactions with the environment. The Offline-RL paradigm is appealing in many real-world applications such as education, healthcare, and robotics, where active data collection is often infeasible (Singh et al., 2022; Singla et al., 2021; Liu et al., 2019). In our work, we develop a method that learns to search in an end-to-end fashion solely from sequences of demonstrations, such as text solutions to reasoning problems and expert trajectories on navigation and games.

## 3 DIFFERENTIABLE TREE SEARCH NETWORK

Differentiable Tree Search Network (D-TSN) is a neural network design that incorporates the algorithmic inductive bias of a best-first search algorithm into the network structure. It learns from sequences of demonstrations to construct search trees by composing submodules, that include an

encoder, a value function, a reward function, and a transition function. If some submodules are readily available, e.g. a world model or simulator, they can be directly used in D-TSN. When these submodules are not available, they can be jointly optimized with the search algorithm. This joint optimization allows the learned imperfect world model to be useful for online search and makes the submodules robust against errors in the world model. We start by describing the version of D-TSN where all submodules are learned.

## 3.1 LEARNABLE SUBMODULES

D-TSN comprises several learnable submodules that function as subroutines in a best-first search algorithm and dynamically construct a computation graph. An illustration is provided in Figure 4. The *Encoder* module ($\mathcal{E}_\theta$) transforms the actual state $s_t$ into a latent state $h_t = \mathcal{E}_\theta(s_t)$, facilitating online search within a latent space. The *Transition* module ($\mathcal{T}_\theta$) approximates the environment's transition function, using $h_t$ and action $a_t$ to predict the subsequent latent state, $h_{t+1} = \mathcal{T}_\theta(h_t, a_t)$, of the transition. The *Reward* module ($\mathcal{R}_\theta$) approximates the environment's reward function, predicting the reward, $r_t = \mathcal{R}_\theta(h_t, a_t)$, for the transition based on $h_t$ and $a_t$. The *Value* module ($\mathcal{V}_\theta$) maps latent state $h_t$ to its estimated state value, $\mathcal{V}_\theta(h_t)$. Dividing the network into these submodules reduces total learnable parameters and injects a strong search inductive bias into the network architecture, preventing overfitting to an arbitrary function that may align with the limited available training data.

## 3.2 TREE SEARCH IN LATENT SPACE

The search begins by encoding the input state $s_0$ into its latent state $h_0$, then proceeds through expansion and backup phases. During expansion, the search tree expands iteratively for a set number of expansion steps, where each step expands a node in the search tree. During backup, Q-values at the root node are recursively computed using the Bellman equation across expanded nodes. Each node represents a latent state reachable from the root, and branches represent actions taken. A candidate set $O$ is maintained during the expansion phase, representing nodes eligible for further expansion. The D-TSN algorithm is detailed in Algorithm 1.

**Expansion Phase** Each search iteration begins by evaluating the path value, $\bar{V}(N)$, of the candidate nodes. The path value is defined as the cumulative sum of rewards from the root node to a particular leaf node $N$, in addition to the value of the leaf node predicted by the value module ($\mathcal{V}_\theta$), i.e.

$$\bar{V}(N) = \mathcal{R}_\theta(h_0, a_0) + ... + \mathcal{V}_\theta(h_N) \tag{1}$$

A naive implementation of the search selects the node $\hat{N}$ with the highest total path value for expansion; however, for a differentiable search, the node $\hat{N}$ is sampled from the candidates using a distribution constructed by applying softmax over the path values of the candidates. Expansion of node $\hat{N}$ is carried out by simulating every action on the node using the transition module ($\mathcal{T}_\theta$). Simultaneously, the associated reward, $\mathcal{R}_\theta(h_{\hat{N}}, a)$, is also computed. The resulting latent states are added to the tree as children of node $\hat{N}$. Additionally, they are added to the candidate set $O$ for subsequent expansions, while $\hat{N}$ is excluded from the set. This can be represented as:

$$O \leftarrow O \cup \{h_a | \, h_a = \mathcal{T}_\theta(h_{\hat{N}}, a); \forall a \in A\} - \hat{N} \tag{2}$$

**Backup Phase** The expansion phase is followed by the *Backup phase*. In this phase, values of all tree nodes are recursively updated using the Bellman equation as follows:

$$Q(N, a) = \mathcal{R}_\theta(h_N, a) + V(N'), \quad \text{where } N' = \mathcal{T}_\theta(h_N, a) \tag{3}$$

$$V(N') = \begin{cases} \mathcal{V}_\theta(h_{N'}), & N' \text{ is a leaf node} \\ \max_a Q(N', a), & \text{otherwise} \end{cases} \tag{4}$$

After the backup phase, Q-values at the root node are returned as the final output of the online search. An illustration of the constructing the tree is shown in Figure 1.

## 3.3 CONSTRUCTION OF COMPUTATION GRAPH

Throughout the expansion and backup phases, illustrated in Appendix Figures 5 and 6 respectively, a dynamic computation graph is constructed where the output Q-values depend on the combination of all the submodules, i.e. Encoder, Transition, Reward, and Value modules. During training, the output

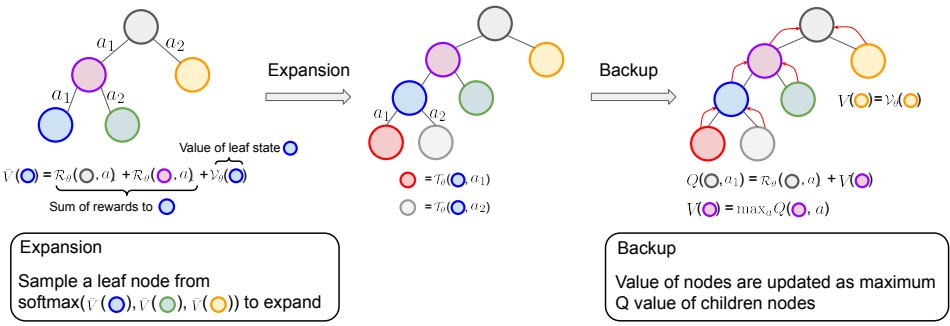

Figure 1: An illustration of the stochastic tree expansion phase and backup phase.

Q-function is evaluated by a loss function and is optimized using gradient-based optimizers such as Stochastic Gradient Descent (SGD). These optimizers backpropagate the gradient of this loss through the entire computation graph and update the parameters of submodules in an end-to-end process. An illustration of the process is provided in Appendix Figure 7.

## 3.4 DISCONTINUITY OF THE LOSS FUNCTION

A pivotal aspect of optimizing D-TSN's parameters through gradient descent is ensuring that the loss function applied to the output Q-values is *continuous in the network's parameter space*. We initiate this discussion by showing that the loss function, when applied to the Q-values computed by expanding a search tree, is continuous in the parameter space, provided that the search tree has a fixed structure.

**Theorem 3.1.** *Given a set of parameterized modules that are continuous in the parameter space $\theta$, the Q-values computed by expanding a search tree with a fixed structure, through the composition of these modules and backpropagation of child values using addition and max operations, are continuous in the parameter space $\theta$. When the tree structure is not fixed, the continuity of the Q-values is not guaranteed. (For a detailed proof, refer to Appendix B.)*

Theorem 3.1 establishes that the Q-values calculated by a fixed-structure tree expansion are continuous in the network's parameter space. The theorem applies to TreeQN (Farquhar et al., 2018), which performs a full tree expansion to a fixed depth, and consequently, the loss function applied to its output Q-values is also continuous, contributing to TreeQN's success in gradient-based optimization.

However, in the case of naive implementation of D-TSN, as outlined in Section 3.2, the search tree is constructed by expanding only those paths that are likely to represent the optimal trajectory from the root node. This results in the output Q-values and the corresponding loss function being dependent on the specific tree, $\tau$, constructed during the search:

$$L(s, a) = \mathcal{L}(Q_\theta(s, a | \tau)) \tag{5}$$

When the network parameters are changed slightly, this naive implementation could generate a different tree structure, which, in turn, would induce a large change in the loss function and affect its continuity in the parameter space.

## 3.5 ENABLING CONTINUITY IN THE LOSS FUNCTION

To overcome the discontinuity issue observed in the naive implementation of D-TSN, we employ a stochastic tree expansion policy. This approach allows us to optimize the expectation of the loss function, defined as:

$$L(s, a) = \mathbb{E}_\tau \Big[ \mathcal{L}(Q_\theta(s, a | \tau)) \Big] = \sum_\tau \pi_\theta(\tau) \mathcal{L}(Q_\theta(s, a | \tau)) \tag{6}$$

The expected loss in Equation (6) is continuous in the parameter space $\theta$ and can be optimized using gradient-based optimization techniques.

## 3.6 STOCHASTIC TREE EXPANSION POLICY

In order to compute the expected loss in Equation (6), let us represent a partial search tree after $t$ iterations as $\tau_t$. The output Q-values, denoted as $Q_\theta(s, a | \tau)$, depend on the final tree $\tau$ sampled after

$T$ iterations. We can define a stochastic tree expansion policy $\pi_\theta(\tau_t)$ that takes a tree $\tau_t$ as input and outputs a distribution over the candidate nodes, facilitating stochastic selection of the node for further expansion and generating the tree $\tau_{t+1}$. We compute the stochastic tree expansion policy by taking softmax over the path value, as defined in Equation (1), of each candidate node as follows:

$$\pi_\theta(n|\tau_t) = \frac{\exp(\bar{V}(n))}{\sum_{n' \in O(\tau_t)} \exp(\bar{V}(n'))} \tag{7}$$

The gradient of the expected loss (Schulman et al., 2015) in Equation (6) can be computed as follows (See Appendix C for the derivation):

$$\nabla_\theta L(s,a) = \mathbb{E}_\tau \Big[ \mathcal{L}(Q_\theta(s,a|\tau)) \sum_{t=1}^{T} \nabla_\theta \log \pi_\theta(n_t|\tau_t) + \nabla_\theta \mathcal{L}(Q_\theta(s,a|\tau)) \Big] \tag{8}$$

### 3.7 REDUCING VARIANCE USING TELESCOPIC SUM

The REINFORCE term of the gradient in Equation (8) usually has high variance due to the difficulty of credit assignment (Pignatelli et al., 2024) in a reinforcement learning type objective; the second part of the gradient equation is the usual optimization of a loss function, so we expect it to be reasonably well behaved. To reduce the variance of the first part of the gradient, we take inspiration from the telescoping sum trick used in Guez et al. (2018).

Let us denote the loss after $t$ iterations as $L_t = \mathcal{L}(Q_\theta(s,a|\tau_t))$. The objective is to minimize the loss after $T$ iterations, represented as $L_T$. Assuming that $L_0 = 0$, we can rewrite $L_T$ as a telescoping sum:

$$L_T = L_T - L_0 = \sum_{t=1}^{T} L_t - L_{t-1} \tag{9}$$

Now, we define a reward term, $r_t$, for selecting node $n$ during the $t^{th}$ iteration as the reduction in the loss value after the $t^{th}$ iteration, i.e. $r_t = L_t - L_{t-1}$. Further, let us represent the return (or rewards-to-go) from iteration $t$ to the final iteration $T$ as $R_t$, which can be computed as:

$$R_t = \sum_{i=t}^{T} r_i = L_T - L_{t-1} \tag{10}$$

Given this, the REINFORCE term from Equation (8) can be reformulated as $\sum_t^T \nabla_\theta \log \pi_\theta(n_t|\tau_t) R_t$, where $L_T$ serves as a baseline that helps in reducing variance. Consequently, the final gradient estimate of the loss in Equation (6) is expressed as:

$$\nabla_\theta L(s,a) = \mathbb{E}_\tau \Big[ \sum_t^T \nabla_\theta \log \pi_\theta(n_t|\tau_t) R_t + \nabla_\theta \mathcal{L}(Q_\theta(s,a|\tau)) \Big] \tag{11}$$

For empirical evaluations, we use a single sample estimate Schulman et al. (2015) of the expected gradient in Equation (11).

### 3.8 LOSS FUNCTIONS

In this section, we define a series of loss functions that are used to train D-TSN. Given a dataset of trajectories where each trajectory is denoted as $\{(s_i,\ a_i,\ r_i,\ Q_i)\}_{i=0}^{T}$, the primary objective is to make the computed Q-values closely approximate the observed Q-values for corresponding states and actions. To achieve this, we minimize the mean squared error between the predicted and observed Q-values. This loss, denoted as $\mathcal{L}_Q$, is defined as:

$$\mathcal{L}_Q = \mathbb{E}_{(s,a,Q) \sim \mathcal{D}}(Q_\theta(s,\ a) - Q)^2 \tag{12}$$

To avoid Q-values for out-of-distribution actions getting overestimated, we incorporate the CQL (Kumar et al., 2020) loss, which encourages the agent to adhere to actions observed within the training data distribution. This loss, $\mathcal{L}_\mathcal{D}$, is defined as:

$$\mathcal{L}_\mathcal{D} = \mathbb{E}_{(s,a) \sim \mathcal{D}} \Big( \log \sum_{a'} \exp(Q_\theta(s,\ a')) - Q_\theta(s,\ a) \Big) \tag{13}$$

Additionally, we incorporate self-supervised consistency loss functions (Schwarzer et al., 2021; Ye et al., 2021) to ensure consistency in the transition and reward networks. Consider actual states $s_i$ and $s_{i+1}$, where $s_{i+1}$ is obtained by taking action $a_i$ in state $s_i$. Their corresponding latent state representations are denoted as $h_i$ and $h_{i+1}$. Here, $h_i = \mathcal{E}_\theta(s_i)$ and $h_{i+1} = \mathcal{E}_\theta(s_{i+1})$. Now, we can use the transition module to predict another latent representation of state $s_{i+1}$, represented as $\bar{h}_{i+1} = \mathcal{T}_\theta(h_i, a_i)$. To ensure that the transition function, $\mathcal{T}_\theta$, provides consistent predictions for the transitions in the latent space, we minimize the squared error between the latent representations $h_{i+1}$ and $\bar{h}_{i+1}$.

$$\mathcal{L}_{\mathcal{T}_\theta} = \mathbb{E}_{(s_i,\ a_i,\ s_{i+1}) \sim \mathcal{D}} (\bar{h}_{i+1} - h_{i+1})^2 \tag{14}$$

In a similar vein, we minimize the mean squared error between the predicted reward $\mathcal{R}_\theta(h_i, a_i)$ and the actual reward observed $r_i$ in the training dataset $\mathcal{D}$.

$$\mathcal{L}_{\mathcal{R}_\theta} = \mathbb{E}_{(s_i,\ a_i,\ r_i) \sim \mathcal{D}} (\mathcal{R}_\theta(h_i, a_i) - r_i)^2 \tag{15}$$

Combining Equations (12), (13), (14) and (15), we define the loss function for training D-TSN as

$$\mathcal{L}_{\text{D-TSN}} = \mathbb{E}_\tau \left[ \lambda_1 \mathcal{L}_Q + \lambda_2 \mathcal{L}_\mathcal{D} \right] + \lambda_3 \mathcal{L}_{\mathcal{T}_\theta} + \lambda_4 \mathcal{L}_{\mathcal{R}_\theta} \tag{16}$$

where $\tau$ denotes the search tree. Additional details on these loss functions are presented in Appendix D.4.

## 4 EXPERIMENTS

### 4.1 D-TSN WITH KNOWN WORLD MODEL

We take an initial step towards learning to perform tree search from instruction sequences for LLMs by first demonstrating that LLMs can be effectively fine-tuned when the world model is known. We study Game of 24 (Yao et al., 2023), a reasoning problem where the transition dynamics and rules are well-defined and readily available. The task is that given four cards drawn from a deck of poker, use basic arithmetic operations to combine the card values to reach 24. The problem can be formulated as follows: a state is a list of current numbers, initially 4; the actions involve selecting an operation as well as two numbers from the current numbers; the transition will calculate the operation and update the list of current numbers by removing the selected two and adding the calculated result; the reward is one when 24 is the only number in the list, and zero otherwise. Since language models can directly process token sequences, we do not need an Encoder in this problem. We use a large language model, Llama3-8B (Dubey et al., 2024), with an additional value head as the value function $\mathcal{V}_\theta$.

### 4.1.1 BASELINE

We compare this version of D-TSN with a supervised fine-tuned (SFT) language model. For a fair comparison, the supervised fine-tuned model is also allowed to perform search during inference, where we use the log probability of the model generating the sequence as its value:

$$\mathcal{V}_\theta(s) = \frac{1}{N} \sum_{i=1}^{N-1} \log p(s_{i+1}|s_{1...i}) \tag{17}$$

where $s_i$ are tokens in the sequence, $p(\cdot|\cdot)$ is the conditional probability of generating the next token. The log probability is divided by $N$ to normalize the effect of sequence length.

### 4.1.2 TRAINING DATASET

We collected all valid Game of 24 problems and their solutions through an exhaustive search of all combinations, then randomly selected 530 problems for evaluation. The remaining 527 problems have 16k valid solutions, from which we randomly sampled a subset for training. We format the text solutions as following format: *"[2,3,7,7]->(2\*7=14)->[3,7,14]->(14+7=21)->[3,21]->(3+21=24)->[24]"*, where *[...]* represents current numbers, and *(...)* represents operations. The operations combine two numbers with an operator, resulting in a new number. We train D-TSN to select operands and operators, and use the transition function to handle the computation of the current numbers after the operation, i.e. given a state *"[2,3,7,7]->(2\*"* and an action to select *"7"* as the next operand, the next state given by the transition function is *"[2,3,7,7]->(2\*7=14)->[3,7,14]->("*. This would allow us to focus more on the difficult part of the solution, which is deciding which numbers to use in the operation and which operator to use. Such transition can be easily obtained in practice as we observe that the SFT model never makes mistakes on these positions in its generation.

### 4.1.3 RESULTS

We train D-TSN using 8 search iterations in training and compare the resulting value function with a supervised fine-tuned model, as described in Equation (17), to guide the search during evaluation.

Table 1: Comparison between D-TSN and SFT on Game of 24, with varying amount of training trajectories, using metrics *Success Rate* – whether the generated expression is valid and results in 24. D-TSN is trained and evaluated with 8 search iterations, SFT is evaluated with 8 search iterations.

| Method | #Training Trajectories | | |
|---|---|---|---|
| | 1k | 2k | 10k |
| D-TSN (n_itr=8) | 31.46 | 38.25 | 45.24 |
| SFT+Search | 12.45 | 15.09 | 21.32 |

To compare D-TSN with a supervised fine-tuned model under different amounts of available data, we sample 1k, 2k, and 10k trajectories to train both methods. Table 1 shows that D-TSN can better guide the search during evaluation when compared to the SFT model. On a closer look, we notice that the log-likelihood of the SFT model as a value function often fails to recognize promising states, assigning a lower value to them, even when the states have already reached 24. As a result, tokens chosen at each step often do not lead to a correct solution. In contrast, D-TSN better recognizes these states and assigns high values to them, guiding each step to select tokens that lead to a correct solution.

## 4.2 D-TSN WITH JOINTLY LEARNED WORLD MODEL

Next, we study two problems where we jointly learn a world model in D-TSN. The states in these problems are represented as pixels of images.

**Navigation** This is a 2D grid-based navigation task designed to quantitatively and qualitatively visualize the agent's generalization capabilities. The environment is a $20 \times 20$ grid featuring a central hall. At the start of each episode, a robot is randomly positioned inside this hall, while its destination is set outside. We present two distinct scenarios: one with a single exit and another with two exits from the central hall. Training is done only in the two exit scenarios. The single exit scenario, which requires a longer-horizon planning to reach the goal, is used to test generalization.

**Procgen** Procgen (Cobbe et al., 2019) is a collection of 16 procedurally generated, game-like environments, specifically designed to evaluate an agent's generalization capability, differentiating it from Atari 2600 games (Mnih et al., 2013). Further details on these domains are presented in Appendix D.1.

We follow the Offline-RL paradigm and train D-TSN using sequences of demonstrations. Please refer to Appendix D.2 for a detailed explanation of the learning framework.

### 4.2.1 TRAINING DATASETS

We use a behavior policy, which can be *optimal or sub-optimal*, to collect demonstration sequences for training. An optimal policy generates a dataset with lower noise and a cleaner training signal, leading to a stable learning process. In contrast, a sub-optimal policy produces a noisier dataset, which consequently restricts the quality of the policy that the agent can learn. The choice of the behavior policy depends on domain-specific requirements and the resources available for data collection. We use an optimal behavior policy for Navigation and a sub-optimal policy for Procgen.

The training dataset, $\mathcal{D}$, consists of trajectories generated using the behavior policy $\pi_B$, where each trajectory is defined as a series of $T$ tuples, each comprising the state observed, action taken, reward observed, and the corresponding Q-value of the observed state, denoted as $\{(s_i, a_i, r_i, Q_i)\}_{i=0}^{T}$. The Q-value for state $s_i$ can be computed by adding the rewards obtained in the trajectory from timestep $i$ onwards, i.e. $Q_i = \sum_{j=i}^{T} r_j$. We limit the number of trajectories to 1000 for each domain to evaluate the sample complexity and generalization capabilities of each method.

### 4.2.2 BASELINES

We benchmark D-TSN against the following prominent baselines:

**Model-free Q-network**   This allows us to assess the significance of incorporating inductive biases into the neural network architecture. This model is trained using the loss defined as:

$$\mathcal{L}_{\text{Q-net}} = \lambda_1 \mathcal{L}_Q + \lambda_2 \mathcal{L}_{\mathcal{D}} \tag{18}$$

**Model-based Search**   In this baseline, we utilize the submodules defined for D-TSN, but the world models and the value module are trained independently of each other. During evaluation, we employ the best-first search, akin to D-TSN, utilizing the independently trained modules. Through this baseline, we assess the benefits derived from the joint optimization of the world model and the search algorithm. For our evaluations, we perform 10 search iterations for each input state. To train this model, we compute the Q-value, $Q_\theta$, *without performing the search* and optimize the loss defined as:

$$\mathcal{L}_{\text{Search}} = \lambda_1 \mathcal{L}_Q + \lambda_2 \mathcal{L}_{\mathcal{D}} + \lambda_3 \mathcal{L}_{\mathcal{T}_\theta} + \lambda_4 \mathcal{L}_{\mathcal{R}_\theta} \tag{19}$$

**TreeQN**   This comparison helps highlight the advantages of an advanced search algorithm, used in D-TSN, that can execute a deeper search while maintaining similar computational constraints. For evaluations, we adhere to a depth of 2 for TreeQN, as described in Farquhar et al. (2018), for both Procgen and navigation domains. Notably, the memory footprint would increase exponentially for larger depth, which prohibits us to use a larger depth. This model is trained using the loss (as discussed in Farquhar et al. (2018)) defined as:

$$\mathcal{L}_{\text{TreeQN}} = \lambda_1 \mathcal{L}_Q + \lambda_2 \mathcal{L}_{\mathcal{D}} + \lambda_3 \mathcal{L}_{\mathcal{R}_\theta} \tag{20}$$

Every method is trained with the same datasets using their respective loss functions. A more comprehensive discussion of the baselines and implementation details can be found in Appendix D.6.

Table 2: Comparison of D-TSN with the baselines on Navigation using metrics *Success Rate* and *Collision Rate*.

| Solver | Success Rate | Absolute Collision Rate |
|---|---|---|
| | *Navigation (2 exits)* | |
| Model-free Q-network | 94.5% (± 0.2%) | 4.4% |
| Model-based Search | 93.2% (± 0.3%) | 6.7% |
| TreeQN | 95.4% (± 0.2%) | 3.8% |
| D-TSN | **99.0%** (± 0.1%) | **0.7%** |
| | *Navigation (1 exit)* | |
| Model-free Q-network | 47.1% (± 0.5%) | 50.2% |
| Model-based Search | 86.9% (± 0.3%) | 12.4% |
| TreeQN | 51.8% (± 0.5%) | 39.2% |
| D-TSN | **99.3%** (± 0.1%) | **0.2%** |

Table 3: Comparison of D-TSN with the baselines on Procgen using metrics *Mean Scores* and *Mean Z-score*. Standard error intervals are reported, based on 1000 evaluation runs per game.

| Games | Model-free Q-network | Model-based Search | TreeQN | **D-TSN** |
|---|---|---|---|---|
| bigfish | 21.67±0.53 | 16.36±0.50 | 20.38±0.52 | **21.76±0.53** |
| bossfight | **8.83±0.18** | 7.33±0.19 | 8.35±0.19 | 8.50±0.19 |
| caveflyer | 2.05±0.12 | 3.52±0.15 | 3.57±0.15 | **3.71±0.15** |
| chaser | 5.77±0.16 | 5.31±0.15 | 6.37±0.17 | **6.77±0.17** |
| climber | 2.85±0.15 | 4.93±0.17 | 4.13±0.16 | **5.62±0.18** |
| coinrun | 5.18±0.16 | 6.33±0.15 | 5.01±0.16 | **6.40±0.15** |
| dodgeball | 1.05±0.06 | 4.26±0.17 | 4.70±0.18 | **5.30±0.19** |
| fruitbot | 14.09±0.37 | 10.76±0.36 | **15.43±0.35** | 14.00±0.37 |
| heist | 0.87±0.09 | 2.24±0.13 | 1.91±0.12 | **2.43±0.14** |
| jumper | 3.21±0.15 | 3.97±0.15 | 3.74±0.15 | **4.65±0.16** |
| leaper | 7.78±0.13 | 6.36±0.15 | 7.85±0.13 | **8.20±0.12** |
| maze | 2.02±0.13 | 2.26±0.13 | 2.50±0.14 | **3.30±0.15** |
| miner | 1.41±0.02 | 1.35±0.05 | **2.06±0.06** | 1.71±0.06 |
| ninja | 5.07±0.16 | 5.07±0.16 | 5.05±0.16 | **6.61±0.15** |
| plunder | **13.77±0.36** | 9.74±0.32 | 12.47±0.35 | 12.75±0.35 |
| starpilot | 15.74±0.39 | 14.83±0.38 | **17.91±0.42** | 16.42±0.41 |
| *Mean Z-Score* | - | 0.10 | 0.27 | **0.35** |

### 4.2.3   RESULTS

**Navigation**   We compare D-TSN against the baselines using success rate and collision rate as evaluation metrics, where success rate refers to the fraction of test levels completed by the agent, and collision rate refers to the fraction of levels failed due to collision with a wall. Results are shown in Table 2.

We observe that D-TSN outperforms the baselines on both navigation scenarios. Notably, when agents are trained on data from the 2-exit scenarios but are tested in the 1-exit scenario, D-TSN, with its powerful inductive bias, retains its performance. In stark contrast, the Model-free Q-network and TreeQN experience a substantial performance decline, which underscores their limited generalization ability. Model-based Search also registers a minor decrease in the success rate, reinforcing the importance of jointly optimizing the world model for enhanced robustness.

**Procgen**   We evaluate the performance of D-TSN and the baselines on all Procgen suite games, comparing mean Z-score and head-to-head wins. We also report the mean scores across 1000 episodes obtained by these methods in Table 3. As Procgen games have different scales, we use model-free Q-network as baseline to compute a normalized score[1], Z-score $= (\mu_\pi - \mu_B)/\sigma_B$, where $\mu_\pi$ and $\mu_B$

---

[1]We use Z-score because the baseline normalized score used in prior works Badia et al. (2020); Kaiser et al. (2020); Mittal et al. (2023) is problematic in our experiments, as described in Appendix D.7.

represent the mean scores obtained by the agent policy and the baseline policy respectively and $\sigma_B$ represents the standard deviation of the scores obtained by the baseline policy.

It is important to note that the training data for Procgen was generated using a sub-optimal behavior policy, leading to noisy training signals in comparison to the relatively noise-free data in the Navigation domain. Despite this, D-TSN reports a higher mean Z-score, averaged across the 16 Procgen games, particularly in climber, coinrun, jumper, and ninja, which require long-term planning. This underscores the stronger inductive bias of D-TSN. In Procgen, Model-based Search lags behind both TreeQN and D-TSN, highlighting that for complex environments, joint optimization enables learning a robust world model. In the head-to-head comparison listed in Table 4, D-TSN won in 13 games against the Model-free Q-network, 16 games against Model-based Search, and 13 games against TreeQN.

Table 4: Head-to-head comparison of D-TSN with the baselines on Procgen using metric *number of games won*.

| Baseline | Tree Search Network (Games won / Total games) |
|---|---|
| Model-free Q-network | 13 / 16 |
| Model-based Search | 16 / 16 |
| TreeQN | 13 / 16 |

Table 5: Comparison of D-TSN with its variants to evaluate the role of *Auxiliary losses*, *REINFORCE* term and *Telescoping Sum*.

| Solver | *Navigation* (1-exit) | *Procgen* |
|---|---|---|
| | Success Rate | Mean Z-score |
| **D-TSN** | **99.3%** (± 0.1%) | **0.31** |
| *w/o Telescoping Sum* | 98.5% (± 0.1%) | 0.28 |
| *w/o REINFORCE term* | 97.7% (± 0.2%) | 0.29 |
| *w/o Auxiliary losses* | 91.1% (± 0.3%) | - |

### 4.3 ABLATION STUDIES

#### 4.3.1 THE IMPACT OF THE REINFORCE TERM AND THE TELESCOPING SUM TRICK

In this study, we assess the impact of both the REINFORCE term and the Telescoping Sum trick on D-TSN's performance. The results, presented in Table 5, show notable differences. Without the telescoping sum, D-TSN sees a modest decrease in the success rate for Navigation (1-exit), moving from 99.3% to 98.5%. Similarly, the mean Z-score for Procgen dips to 0.28. The omission of the REINFORCE term also marks a decline, with the Navigation (1-exit) success rate landing at 97.7% and the mean Z-score for Procgen dipping to 0.29.

#### 4.3.2 THE CONTRIBUTION OF AUXILIARY LOSSES

In this study, we explore the contribution of auxiliary losses to D-TSN's performance. As outlined in Table 5, the absence of these auxiliary losses leads to a more pronounced decline in the performance. Specifically, the success rate in the Navigation (1-exit) task drops significantly to 91.1% compared to 99.3% with auxiliary losses.

#### 4.3.3 ENHANCING PERFORMANCE THROUGH DEEPER SEARCH

In this study, we explore the benefits of deeper search in the D-TSN framework by increasing search iterations in Navigation and Game of 24 problems. For Navigation, we maintain an identical number of iterations in the training and evaluation phases to prevent any distribution shifts in the world model. For Game of 24, we can use a larger search iteration during evaluation, as the world model remains fixed.

Table 6: Comparison of D-TSN trained with different number of *search iterations* ('n_itr') to evaluate the performance gain by performing deeper searches.

| Solver | *Navigation* (2-exits) | *Navigation* (1-exit) |
|---|---|---|
| D-TSN (n_itr=5) | 97.4% | 96.6% |
| D-TSN (n_itr=10) | 99.0% | 99.3% |
| D-TSN (n_itr=20) | 99.5% | 99.1% |

Table 7: Comparison of D-TSN using different number of search iterations during evaluation. All methods are trained with 2k trajectories.

| | #Search iterations in evaluation | | | | |
|---|---|---|---|---|---|
| | 1 | 4 | 8 | 16 | 128 |
| D-TSN (n_itr=1) | 20.75 | 31.51 | 38.16 | 49.76 | 89.29 |
| D-TSN (n_itr=4) | | 32.22 | 41.13 | 49.67 | 90.24 |
| D-TSN (n_itr=8) | | | 38.25 | 49.91 | 91.42 |
| D-TSN (n_itr=16) | | | | 49.58 | 92.38 |

The results, as listed in Table 6 and Table 7 indicate that a higher number of iterations leads to an improvement in the success rate in both Navigation and Game of 24. For Navigation, where the world model is not known, the improvement suggests that D-TSN is able to exploit the jointly learned world model effectively, and can potentially be scaled up further given additional computational resources. For Game of 24, we see performance consistently improve as the number of search iterations during evaluation increases, highlighting the importance of search. Notably, the performance reaches around 90% for D-TSN when there are sufficient search iterations during evaluation, suggesting that the value function learned by D-TSN can recognize the promising states and guide the generation to those states. We also observe that the number of search iterations performed during training does not affect the performance much in Game of 24. One explanation is that we are using a ground-truth world model, and performing a search using actual states. The REINFORCE term in Equation (11) would act similarly to the CQL loss in Equation (13). Learning a world model jointly in D-TSN is also possible in principle in language model tasks, but scaling considerations become important to manage the large action space and observation space. We leave this exploration for future works.

### 4.3.4 ROBUSTNESS OF THE WORLD MODEL

A key challenge in employing learned world models for search is addressing the compounding errors, which impact the accuracy and effectiveness of search. This study shows that the joint optimization of both the world model and the search algorithm compensates for these inaccuracies, ensuring the world model is usable in deeper online searches.

Table 8: Comparison of D-TSN and Model-based Search to highlight the robustness of the world model when performing deeper searches ('n_itr' refers to number of search iterations).

| Solver | Navigation (2-exits) | Navigation (1-exit) |
|---|---|---|
| D-TSN (n_itr=10) | 99.0% | 99.3% |
| D-TSN (n_itr=20) | 99.3% | 99.4% |
| D-TSN (n_itr=50) | 99.7% | 99.6% |
| Model-based Search (n_itr=10) | 93.2% | 86.9% |
| Model-based Search (n_itr=20) | 91.1% | 84.6% |
| Model-based Search (n_itr=50) | 89.5% | 80.4% |

As demonstrated in Table 8, the world model trained jointly with the search algorithm consistently outperforms the independently trained model, especially as the number of iterations increases.

## 5 CONCLUSION

In this paper, we introduce Differentiable Tree Search Network (D-TSN), a novel framework that learns to conduct differentiable tree search from just sequences of demonstrations. D-TSN can leverage a known world model, or jointly learn a world model end-to-end with other search submodules. D-TSN conducts a best-first style search, which allows it to search deeper and explore more promising states. A naive incorporation of best-first search could lead to discontinuity of the loss in the parameter space, an issue we address by employing a stochastic tree expansion policy. We optimize the expected loss function using a REINFORCE-style objective and propose a telescoping sum trick to reduce the variance of the gradient for this expected loss. We evaluate D-TSN in two scenarios: 1) when the world model is known – we experiment with a language model reasoning task, Game of 24; 2) when the world model is not known – we experiment with 2D grid navigation and Procgen games. Through our experiments, we show that D-TSN is effective, outperforming baselines in both scenarios, especially when the world model is jointly learned.

Nonetheless, the strength of the current implementation of D-TSN is currently limited to deterministic decision-making problems with a discrete action space. We have demonstrated the method on latent world models with 2D visual observations but have yet to do so on language observations. To cater to a broader spectrum of decision-making problems, there is also a need to revamp the transition model to manage stochastic world scenarios, which include common usage of language and visual language models. We aim to address these problems in our future works.

### ACKNOWLEDGMENTS

This research is supported by the Ministry of Education, Singapore, under its Academic Research Fund Tier 1 (A-8001814-00-00).

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

## A    DIFFERENTIABLE TREE SEARCH NETWORK ALGORITHM

### A.1    DIFFERENTIABLE TREE SEARCH NETWORK PSEUDO-CODE

---

**Algorithm 1:** Differentiable Tree Search (D-TSN)

---

**Input:** Input state, $s_{root}$
**Result:** Q-values, $Q(s_{root}, a)$

$h_{root} \leftarrow \mathcal{E}_\theta(s_{root})$ ;                          //Encode $s_0$ to its latent state $h_0$
$node_{root} \leftarrow initialise(h_{root})$ ;                          //Initialize root node
$Open \leftarrow \{node_{root}\}$ ;                          //Initialize candidate set $Open$
// **Expansion phase**
$itr \leftarrow 0$;
**repeat**
   **foreach** $node \in Open$ **do**
      $h_{node} \leftarrow getLatent(node)$ ;                          //Get latent state of $node$
      $\bar{V}(node) \leftarrow sumOfRewards(node) + \mathcal{V}_\theta(h_{node})$ ;    //Compute path values
   $\pi_{tree} \leftarrow \text{softmax}_n\left(\bar{V}(n)\right)$ ;              //Compute the tree expansion policy
   $node^* \leftarrow \text{sample}(\pi_{tree})$ ;                          //Sample the node to expand
   **foreach** $a \in Actions$*;* **do**
      $h_{child} \leftarrow T_\theta(h_{node^*}, a)$ ;              //Compute the next latent state
      $r_{child} \leftarrow \mathcal{R}_\theta(h_{node^*}, a)$ ;         //Compute reward for the transition
      $createNode(h_{child}, r_{child})$ ;                          //Create the child node
   $Open \leftarrow Open \cup \{child_a | child_a = getChild(node^*, a); \forall a \in A\} - node^*$ ; //Update
   the open set
   $itr \leftarrow itr + 1$;
**until** $itr < MAX\_ITR$;
// **Backup phase**
**foreach** $node \in Tree$*, iterating from leaf nodes to the root node;* **do**
   **if** *node is a leaf;* **then**
      $h_{node} \leftarrow getLatent(node)$ ;                          //Get latent state of $node$
      $V(node) = \mathcal{V}_\theta(h_{node})$ ;              //Compute value using Value module
   **else**
      **foreach** $a \in Actions$*;* **do**
         $node_{child[a]} \leftarrow getChild(node, a)$ ;         //Get child of $node$ that
         corresponds to action $a$
         $h_{node} \leftarrow getLatent(node)$ ;                          //Get latent state of $node$
         $r_{node} \leftarrow \mathcal{R}_\theta(h_{node}, a)$ ;         //Get reward using Reward module
         $Q(node, a) \leftarrow r_{node} + V\left(node_{child[a]}\right)$
      $V(node) \leftarrow \max_a Q(node, a)$
**return** $Q(node_{root})$ ;                          //Return Q-value of the root node

---

### A.2    PSEUDO-CODE EXPLANATION

D-TSN initializes the search by converting the input state to its latent representation and proceeds in two phases: Expansion and Backup as detailed in the following sections.

#### A.2.1    INITIALIZATION

Given an input state $s_{\text{root}}$, the algorithm begins by encoding this state into its latent representation $h_{\text{root}}$ using an encoder $\mathcal{E}_\theta$. This latent representation serves as the root node of the search tree.

### A.2.2 EXPANSION PHASE

- The algorithm initiates a set of candidate nodes, termed 'Open', starting with the root node.
- In each iteration, the algorithm considers every node in the 'Open' set, retrieves its latent state, and computes an interim value $\bar{V}(\text{node})$, which combines the cumulative rewards of the node's path with a value estimation from the value module $\mathcal{V}_\theta$.
- Using the path values of nodes in 'Open', the tree expansion policy, $\pi_{\text{tree}}$, is computed. From this policy, a node, $node^*$, is sampled for expansion.
- Each possible action from $node^*$ results in the creation of a child node. This is achieved by leveraging the transition module $\mathcal{T}_\theta$ to predict the latent state of the child and the reward module $\mathcal{R}_\theta$ to determine the associated reward. After expansion, $node^*$ is removed from 'Open' and its children are added.
- This expansion process continues until a predetermined number of iterations, MAX_ITR, is reached.

### A.2.3 BACKUP PHASE

- Starting from the leaf nodes, the algorithm backpropagates value estimates to the root.
- For each leaf node, its value is directly computed from the value module $\mathcal{V}_\theta$. For non-leaf nodes, the Q-value for each action is estimated by combining the reward for that action with the value of the corresponding child node.
- The value of a non-leaf node is set to the maximum Q-value among its actions.

### A.2.4 OUTPUT

The algorithm finally returns the Q-values associated with the root node, $Q(node_{\text{root}})$, providing an estimation of the value of taking each action from the initial state.

This explanation provides a high-level view of the D-TSN algorithm's operation. By breaking down the search process into expansion and backup phases, the pseudo-code highlights how D-TSN incrementally builds the search tree and then consolidates value estimates back to the root.

## B CONTINUITY OF THE LOSS FUNCTION

In this section, we show that the Q-function represented by a search tree with fixed structure is continuous in network's parameter space.

Suppose we have two functions $f(x)$ and $g(x)$ which are continuous at any point $c$ in their domains.

**Lemma B.1.** *Continuity of Composition: The composition of two continuous functions, denoted as $f(g(x))$, retains continuity. (Theorem 4.7 in Rudin (1976))*

**Lemma B.2.** *Continuity of Sum operation: The result of adding two continuous functions, expressed as $f(x) + g(x)$, is a continuous function. (Theorem 4.9 in Rudin (1976))*

**Lemma B.3.** *Continuity of Max operation: Applying Max over two continuous functions, expressed as $\max(f(x), g(x))$, results in a function that is continuous.*

*Proof.* Consider a function $h(x) = \max(f(x), g(x))$, and we aim to demonstrate that $h(x)$ is continuous. Now, $h(x)$ can be expressed as a combination of continuous functions:

$$h(x) = \frac{f(x) + g(x) + |f(x) - g(x)|}{2}$$

Since sums and absolute values of continuous functions are continuous (Rudin, 1976), $h(x)$ is continuous. Hence, the maximum of two continuous functions is also a continuous function. □

Rewriting the Theorem 3.1 with proof below:

**Theorem B.4.** *Given a set of parameterised modules that are continuous in the parameter space $\theta$, the Q-function computed by expanding a fixed search tree by composing these modules and backpropagating the children values using addition and max operations is continuous in the parameter space $\theta$. When the tree structure is not fixed, the continuity of the Q-values is not guaranteed.*

*Proof.* Let us represent the set of parameters, encoder module, transition module, reward module, and value module respectively as $\theta$, $\mathcal{E}_\theta$, $\mathcal{T}_\theta$, $\mathcal{R}_\theta$, and $\mathcal{V}_\theta$. These submodules are assumed to be composed of simple neural network architectures, comprising linear and convolutional layers, with the Rectified Linear Unit (ReLU) serving as the activation function. These submodules, therefore, are continuous within the parameter space.

We can subsequently rewrite the Q-value as the output of a full tree expansion as follows:

$$Q(s_0, a_0) = Q(h_0, a_0) \tag{21}$$
$$= r_0 + V(h_1)$$

where,

$$h_0 = \mathcal{E}_\theta(s_0) \tag{22}$$
$$r_t = \mathcal{R}_\theta(h_t, a_t) \tag{23}$$
$$h_{t+1} = \mathcal{T}_\theta(h_t, a_t) \tag{24}$$
$$Q(h_t, a_t) = \mathcal{R}_\theta(h_t, a_t) + V(h_{t+1}) \tag{25}$$
$$V(h_t) = \begin{cases} \mathcal{V}_\theta(h_t) & \text{if } h_t \text{ is a leaf} \\ \max_a \left( Q(h_t, a) \right) & \text{otherwise} \end{cases} \tag{26}$$

Given that $\mathcal{E}_\theta$, $\mathcal{R}_\theta$, and $\mathcal{T}_\theta$ are continuous, $h_0$, $r_t$, and $h_{t+1}$ in Equations (22) and (24) are similarly continuous (derived from Lemma B.1).

Further, $V(h_t)$ in Equation (26) can either be $\mathcal{V}_\theta(h_t)$, if $h_t$ is a leaf node, or $\max_a(Q(h_t, a))$ otherwise. In the first scenario, $V(h_t)$ retains continuity by assumption. In the second scenario, if $Q(h_t, a_t)$ is continuous, then $V(h_t)$ remains continuous as per Lemma B.3.

Now, we show that $Q(h_t, a_t)$ is continuous using a recursive argument that for any node in the search tree, if the Q-values of all its child nodes are continuous, then its Q-value is also continuous. Q-value of an internal tree node $h_t$ can be written as $Q(h_t, a_t) = \mathcal{R}_\theta(h_t, a_t) + V(h_{t+1})$, where $h_{t+1} = \mathcal{T}_\theta(h_t, a_t)$ is the child node of $h_t$. Considering the base case, when the $h_{t+1}$ is a leaf node, then $Q(h_t, a_t) = \mathcal{R}_\theta(h_t, a_t) + \mathcal{V}_\theta(h_{t+1})$, which is continuous as per Lemma B.2. Consequently, $V(h_t)$ maintains continuity. Applying this logic recursively, the Q-value $Q(h_t, a_t)$ of all the tree nodes maintains continuity.

Thus, we can decompose the Q-value at the root node, $Q(s_0, a_0)$, as a composition of continuous functions, ensuring that the output Q-value, $Q(s_0, a_0)$, is continuous.

Next, we show that the continuity of Q-function is not guaranteed when the structure of the search tree is not fixed by giving an example.

Consider a simple environment with two possible actions $a_1$ and $a_2$, and a state is represented by a 2-D vector. We parameterize the reward, value, transition functions as simple matrices: $R_i \in \mathbb{R}^2$ is a reward function that maps a state to a real number representing the reward for doing action $a_i$ at the state; $V \in \mathbb{R}^2$ is the value function that maps a state to a real number representing the value of a state; $T_i \in \mathbb{R}^{2 \times 2}$ is a transition function that maps a state to a next state after doing action $a_i$. Consider a search tree with root node $s \in \mathbb{R}^2$ and a child node $x \in \mathbb{R}^2$ derived by doing action $a_1$. $x$ has two child node $n_1$ and $n_2$. The path value of $n_1$, defined by Equation (1), is $\bar{V}(n_1) = \mathcal{R}_\theta(s, a_1) + \mathcal{R}_\theta(x, a_1) + \mathcal{V}_\theta(n_1) = R_1^T s + R_1^T x + V^T T_1 x$. Similarly, the path value of $n_2$ is $\bar{V}(n_2) = \mathcal{R}_\theta(s, a_1) + \mathcal{R}_\theta(x, a_2) + \mathcal{V}_\theta(n_2) = R_1^T s + R_2^T x + V^T T_2 x$.

Consider we choose between $n_1$ and $n_2$ for the next expansion of the tree. When $R_1^T x + V^T T_1 x > R_2^T x + V^T T_2 x$, $n_1$ is expanded, two children of $n_1$ has value $R_1^T T_1 x + V^T T_1 T_1 x$ and $R_2^T T_1 x + V^T T_2 T_1 x$, value of $x$ after backup is the maximum of value of $n_2$ and the values of two children of $n_1$, which is $\max\{R_2^T x + V^T T_2 x, \max\{R_1^T T_1 x + V^T T_1 T_1 x, R_2^T T_1 x + V^T T_2 T_1 x, \}\}$.

When $R_1^T x + V^T T_1 x < R_2^T x + V^T T_2 x$, $n_2$ is expanded, two children of $n_2$ has value $R_1^T T_2 x + V^T T_1 T_2 x$ and $R_2^T T_2 x + V^T T_2 T_2 x$, value of $x$ after backup is maximum of value of $n_1$ and the values of two children of $n_2$, which is $\max\{R_1^T x + V^T T_1 x, \max\{R_1^T T_2 x + V^T T_1 T_2 x, R_2^T T_2 x + V^T T_2 T_2 x, \}\}$.

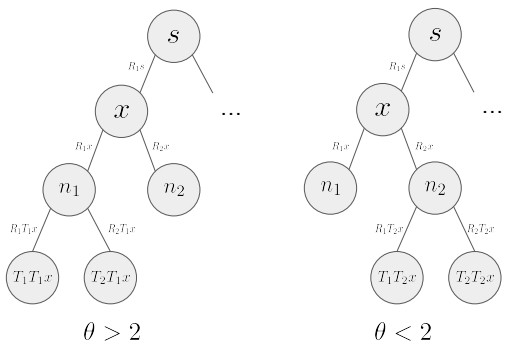

Figure 2: Visualization of a discontinuous example.

Let's assign the following values to the mentioned symbols:

$$x = \begin{bmatrix} 2 \\ 2 \end{bmatrix}; \quad V = \begin{bmatrix} \theta \\ 1 \end{bmatrix}; \quad R_1 = \begin{bmatrix} 0 \\ 1 \end{bmatrix}; \quad R_2 = \begin{bmatrix} 1 \\ 0 \end{bmatrix}; \quad T_1 = \begin{bmatrix} 0 & 2 \\ 1 & 0 \end{bmatrix}; \quad T_2 = \begin{bmatrix} 0 & 1 \\ 1 & 2 \end{bmatrix}$$

We have $R_1^T x + V^T T_1 x = 4\theta + 4$, and $R_2^T x + V^T T_2 x = 2\theta + 8$. When $\theta > 2$, $n_1$ will be expanded, and when $\theta < 2$, $n_2$ will be expanded, see Figure 2 for a visualization of the example. We have

$$\lim_{\theta \to 2^+} Q(s, a_1)$$
$$= R_1^T s + \lim_{\theta \to 2^+} \max(R_2^T x + V^T T_2 x, \max(R_1^T T_1 x + V^T T_1 T_1 x, R_2^T T_1 x + V^T T_2 T_1 x,)) \quad (27)$$
$$= 16$$

$$\lim_{\theta \to 2^-} Q(s, a_1)$$
$$= R_1^T s + \lim_{\theta \to 2^-} \max(R_1^T x + V^T T_1 x, \max(R_1^T T_2 x + V^T T_1 T_2 x, R_2^T T_2 x + V^T T_2 T_2 x,)) \quad (28)$$
$$= 32$$

Combining Equation (27) and Equation (28) we have that $Q(r, a_1)$ is discontinuous at $\theta = 2$. $\qquad \square$

## C  DERIVATION OF THE GRADIENT OF THE EXPECTED LOSS FUNCTION

Let us represent the Q-values predicted by D-TSN as $Q_\theta(s, a|\tau)$, which depends on the final tree $\tau$ sampled after $T$ trials of the online search. Let us denote the corresponding loss function on this output Q-value as $\mathcal{L}\Big(Q_\theta(s, a|\tau)\Big)$. Our objective is to compute the gradient of the expected loss value, averaging over trees sampled.

The gradient of expected loss, considering the expectation over the sampled trees, is derived as:

$$\nabla_\theta L(s,a) = \nabla_\theta \mathbb{E}_\tau \Big[ \mathcal{L}\big(Q_\theta(s,a|\tau)\big) \Big]$$

$$= \nabla_\theta \sum_\tau \pi_\theta(\tau) \mathcal{L}\big(Q_\theta(s,a|\tau)\big)$$

$$= \sum_\tau \nabla_\theta \Big[ \pi_\theta(\tau) \mathcal{L}\big(Q_\theta(s,a|\tau)\big) \Big]$$

$$= \sum_\tau \mathcal{L}\big(Q_\theta(s,a|\tau)\big) \nabla_\theta \pi_\theta(\tau) + \sum_\tau \pi_\theta(\tau) \nabla_\theta \mathcal{L}\big(Q_\theta(s,a|\tau)\big)$$

$$= \sum_\tau \pi_\theta(\tau) \mathcal{L}\big(Q_\theta(s,a|\tau)\big) \nabla_\theta \log \pi_\theta(\tau) + \sum_\tau \pi_\theta(\tau) \nabla_\theta \mathcal{L}\big(Q_\theta(s,a|\tau)\big)$$

$$= \mathbb{E}_\tau \Big[ \mathcal{L}\big(Q_\theta(s,a|\tau)\big) \nabla_\theta \log \pi_\theta(\tau) + \nabla_\theta \mathcal{L}\big(Q_\theta(s,a|\tau)\big) \Big]$$

$$= \mathbb{E}_\tau \Big[ \mathcal{L}\big(Q_\theta(s,a|\tau)\big) \nabla_\theta \log \prod_{t=1}^{T} \pi_\theta(n_t|\tau_t) + \nabla_\theta \mathcal{L}\big(Q_\theta(s,a|\tau)\big) \Big]$$

$$= \mathbb{E}_\tau \Big[ \mathcal{L}\big(Q_\theta(s,a|\tau)\big) \sum_{t=1}^{T} \nabla_\theta \log \pi_\theta(n_t|\tau_t) + \nabla_\theta \mathcal{L}\big(Q_\theta(s,a|\tau)\big) \Big]$$

Leveraging the telescoping sum trick, as elaborated in Section 3.7, the gradient of the expected loss can be rewritten as a lower-variance estimate:

$$\nabla_\theta L(s,a) = \mathbb{E}_\tau \Big[ \sum_{t=1}^{T} \nabla_\theta \log \pi_\theta(n_t|\tau_t) R_t + \nabla_\theta \mathcal{L}\big(Q_\theta(s,a|\tau)\big) \Big]$$

where

$$R_t = \sum_{i=t}^{T} r_i = \mathcal{L}_T - \mathcal{L}_{t-1}$$

$\mathcal{L}_t = \mathcal{L}\big(Q_\theta(s,a|\tau_t)\big)$, representing the loss value after the $t^{th}$ search iteration.

In practice, we utilize the single-sample estimate for the expected gradient, as elaborated in Schulman et al. (2015)

## D    EXPERIMENTS

### D.1    TEST DOMAINS

We use a grid navigation task and the Procgen games to study D-TSN under the scenario when world models are not available. A visualization of these two tasks can be viewed in Figure 3.

### D.1.1    NAVIGATION

The grid navigation task serves as a foundational test that mimics the challenges a robot might face when navigating in a 2D grid environment. This environment provides both a quantitative metric and a qualitative visualization to understand an agent's capacity to generalize its policy. Specifically, this task involves a $20 \times 20$ grid with a central hall. At the beginning of each episode, the robot is positioned at a random point within this central hall. Simultaneously, a goal position is sampled randomly at a location outside the hall, challenging the robot to find its way out and reach this target. There are two variations of this task. The first provides the robot with a single exit from the central hall, while the second offers two exits. The single-exit hall scenario is similar to the two-exit scenario but requires a longer-horizon planning to successfully evade the walls to exit the hall and reach the goal.

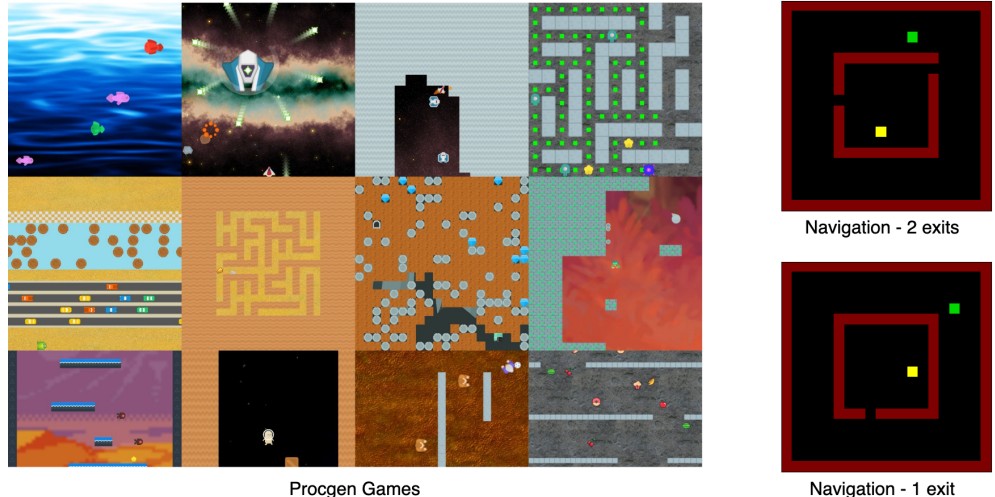

Figure 3: A sample visualization of Procgen games (left) and Grid Navigation (right).

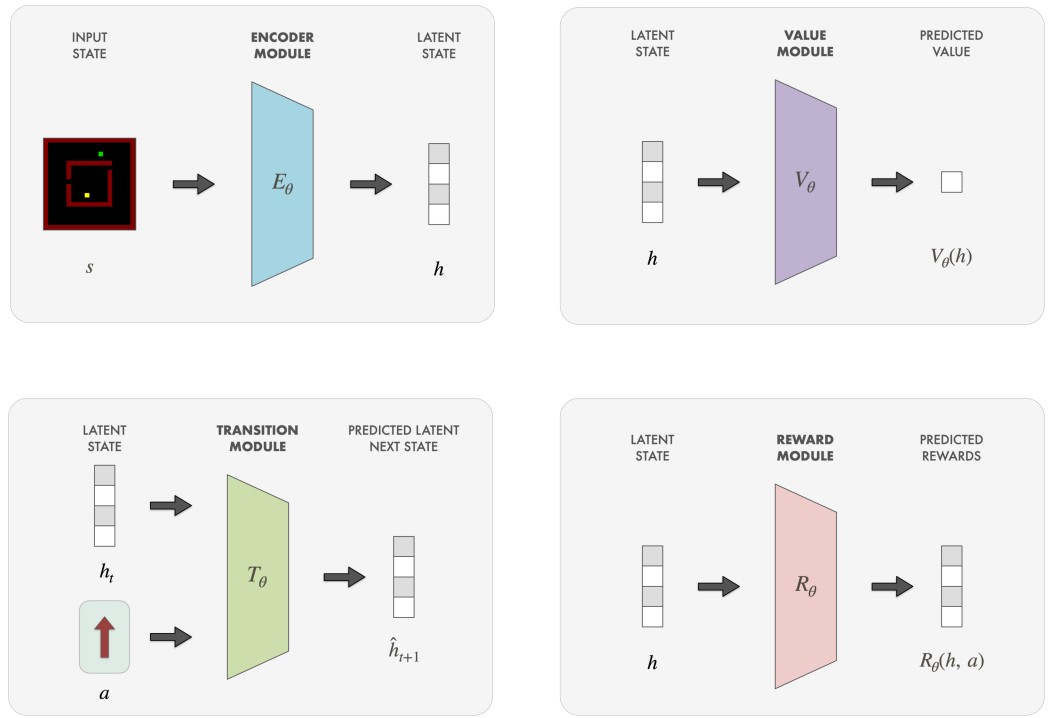

Figure 4: An illustration of the learnable submodules in Differentiable Tree Search Network

### D.1.2 PROCGEN

Procgen is a unique suite consisting of 16 game-like environments, each of which is procedurally generated. This means that they are designed to present slightly altered levels every time they are played. Such design intricacy makes Procgen an ideal choice to test an agent's generalization capabilities. It stands in contrast to other commonly used testing suites, like the renowned Atari 2600 games (Mnih et al., 2013; 2016; Badia et al., 2020). The diverse array of environments within Procgen emphasizes the pivotal role of robust policy learning. The environmental diversity in Procgen

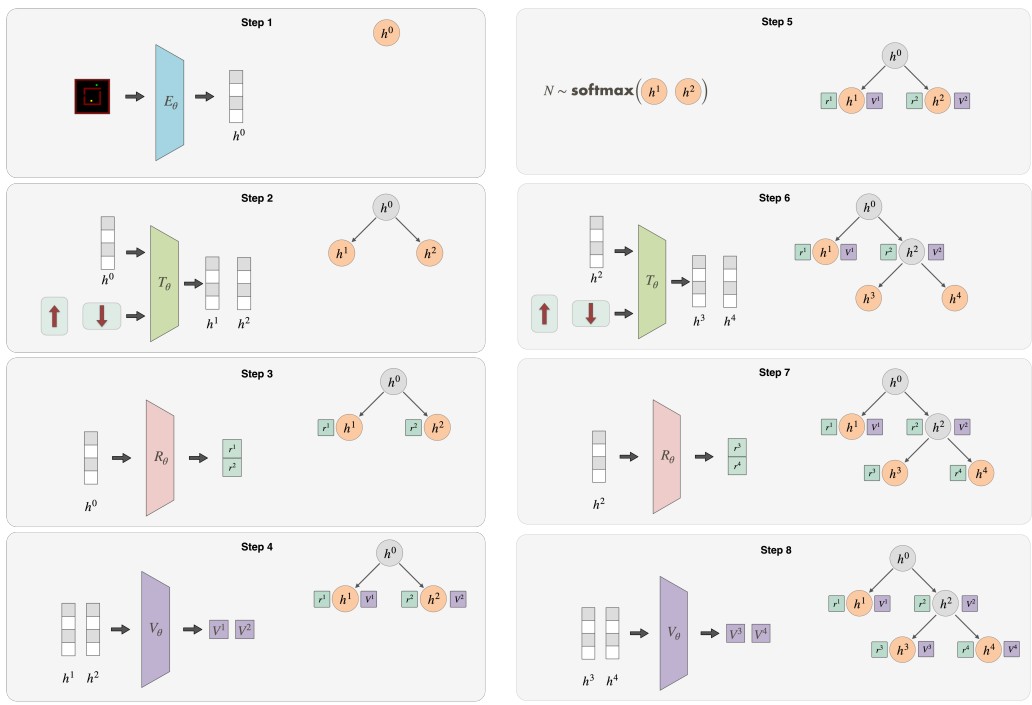

Figure 5: An illustration of the Expansion Phase in  Differentiable Tree Search Network

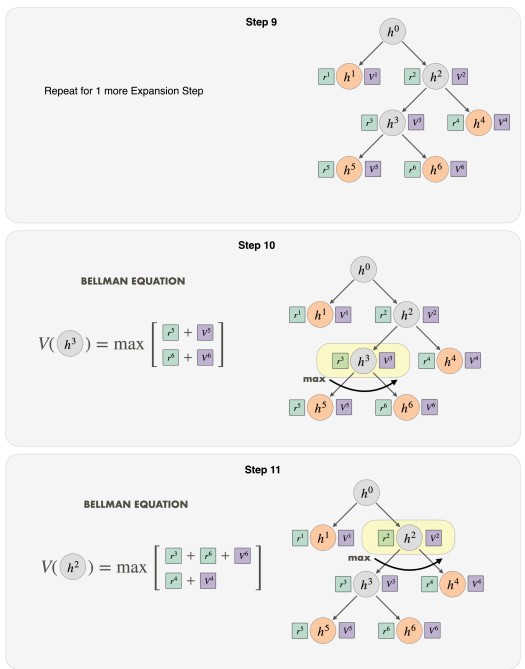

Figure 6: An illustration of the Backup Phase in  Differentiable Tree Search Network

underlines the importance of robust policy learning for successful generalization. The open-source code for the environments is publicly accessible at https://github.com/openai/procgen.

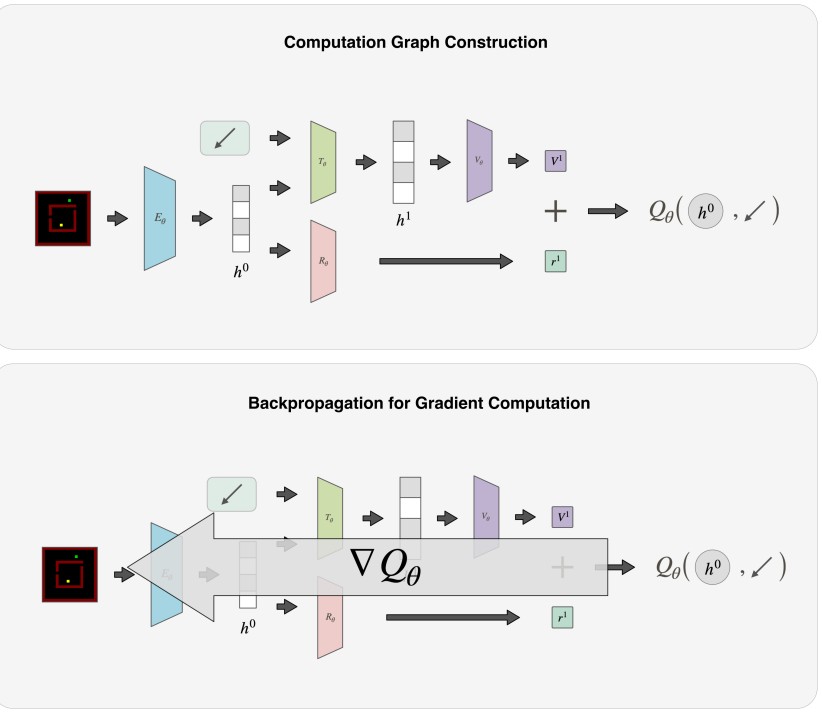

Figure 7: An illustration of the computation graph construction in Differentiable Tree Search Network

## D.2 LEARNING SETUP

D-TSN can serve as a drop-in replacement for popular model-free predictors. It can be trained using both online and offline reinforcement learning algorithms. In this work, we employ the offline reinforcement learning (Offline-RL) framework to focus on the sample complexity and generalization capabilities of D-TSN when compared with the baselines.

Offline-RL, often referred to as batch-RL, is the scenario wherein an agent learns its policy solely from a fixed offline dataset of experiences, without further interactions with the environment. Offline-RL framework poses significant challenges, especially when it comes to the generalization capabilities of methods, even for seemingly straightforward tasks like navigation. Consider, for instance, that the optimal policy is employed to collect experiences from an environment. It would predominantly select the optimal actions at every state. This means that only a fraction of the vast state-action space would be covered in the dataset. As a result, the model might not learn about many interactions, such as what happens when it collides with a wall, due to the limited training data on such environment interactions. When this world model is employed in an online search scenario, the search process might unknowingly venture into out-of-distribution state space. These explorations, stemming from the model's limited generalization capabilities, can lead to overly optimistic value predictions, subsequently affecting the Q-values computation at the root node. In practical terms, the online search might mistakenly believe it can travel through a wall to reach its goal more quickly and would then run into it.

However, the D-TSN design offers a solution to this problem by jointly optimizing both the world model and the online search. During the training phase, when the online search strays into the out-of-distribution states, it might overestimate the value of these states. This overestimation would then influence the final output after the search. When such a mismatch between the post-search output and the expert action is detected, the gradient descent algorithm adjusts these overestimated values, effectively lowering them to align the final Q-values more closely with the expert action. As a direct consequence of this, by the end of the training phase, the search process will have effectively learned to ignore these the out-of-distribution states. Even if such expansion does occur due to $\epsilon$-greedy

exploration during inference, the predicted value for the out-of-distribution state will be small and thus will have negligible impact on the Q-values of the root node due to the max operation during backup.

## D.3 TRAINING DATASETS

We employ a separate behavior policy to collect the offline training dataset. Here, the behavior policy can be *optimal or sub-optimal*. An optimal policy generates a dataset with lower noise and a cleaner training signal, leading to a stable learning process. In contrast, a sub-optimal policy produces a noisier dataset, which consequently restricts the quality of the policy that the agent can learn. The selection of the behavior policy depends on domain-specific requirements and the resources available for data collection. This work explores datasets generated using both optimal (in the Navigation domain) and sub-optimal (in the Procgen domain) behavior policies.

The training dataset, $\mathcal{D}$, consists of trajectories generated using the behavior policy $\pi_B$, where each trajectory, $\tau_i$, is defined as a series of $T$ tuples, each comprising the state observed, action taken, reward observed, and the corresponding Q-value of the observed state, denoted as:

$$\tau_i = \left\{ (s_{t,i}, \ a_{t,i}, \ r_{t,i}, \ Q_{t,i}) \right\}_{t=0}^{T}$$

The Q-value for state $s_{t,i}$ can be computed by adding the rewards obtained in the trajectory from timestep $t$ onwards, i.e.

$$
\begin{aligned}
Q_{t,i} &= Q^{\pi_B}(s_{t,i}) \\
&= r_{t,i} + r_{t+1,i} + r_{t+2,i} + ...r_{T,i} \\
&= \sum_{k=t}^{T} r_{k,i}
\end{aligned}
$$

In order to evaluate the sample complexity and generalization capabilities of each method, we collected a small number of 1000 trajectories for each test domain for our experiments.

### D.3.1 NAVIGATION

For our navigation task, which is relatively small in size, we are able to compute the optimal policy for any given state and configuration. We employ the value iteration algorithm for this purpose, as detailed by Sutton & Barto (2018). At the beginning of each episode, we formulate a random passage through the central hall. Subsequently, we also randomly determine the starting position of the robot within the hall and its goal position outside of it. We collect a total of 1000 expert trajectories for training. Each of these trajectories incorporates a sequence comprising states, actions, rewards, and Q-values observed throughout the episode.

### D.3.2 PROCGEN

When it comes to Procgen, even though there isn't a public repository of pre-trained models, there exists an open-source code base for Phasic Policy Gradient (PPG) (Cobbe et al., 2021). With this in hand, we could effectively train a decent, *but sub-optimal*, policy for every individual Procgen game starting from the ground up. We rely on the default set of hyperparameters for training a specific policy for each game. Using these policies, we gather sample trajectories for 1000 successful episodes. Just like in the case of Navigation, each of these trajectories represents a sequence comprising states, actions, rewards, and Q-values observed throughout the episode.

## D.4 LOSS FUNCTIONS

As mentioned in the previous section, D-TSN can be trained using various online and offline RL methods. In this work, we are focusing on offline-RL framework and use Behavior Cloning to train the parameters of D-TSN. Behavior Cloning is a type of supervised learning where the objective is to make the agent's prediction closely approximate the actions taken by the behavior policy in the states collected in the training dataset. To achieve this, we minimize the mean squared error between the predicted and target Q-values. This loss, denoted as $\mathcal{L}_Q$, is expressed as:

$$\mathcal{L}_Q = \mathbb{E}_{(s_i, \ a_i, \ Q_i) \sim \mathcal{D}} \Big( Q_\theta(s_i, \ a_i) - Q_i \Big)^2 \tag{29}$$

Moreover, during the online search, the transition, reward, and value networks operate on the latent states. Consequently, it is important to ensure that the input to these networks is of a consistent scale,

as suggested in (Schwarzer et al., 2021; Ye et al., 2021). To achieve this, we apply hyperbolic tangent (Tanh) normalization on the latent states, thereby adjusting their scale to fall within the range $(-1, 1)$.

$$h = tanh(x)$$
$$= \frac{e^x - e^{-x}}{e^x + e^{-x}} \quad \in (-1, 1)$$

### D.4.1 AUXILIARY LOSS FOR OUT-OF-DISTRIBUTION ACTIONS

In the offline-RL setting, there's a risk that Q-values for out-of-distribution actions might be overestimated (Kumar et al., 2020) as the behavior policy can only cover a limited part of the state-action distribution. To address this, we incorporate an additional CQL (Kumar et al., 2020) loss which encourages the agent to adhere to actions observed within the training data distribution. This loss, $\mathcal{L}_{\mathcal{D}}$, is defined as:

$$\mathcal{L}_{\mathcal{D}} = \mathbb{E}_{(s_i,\, a_i) \sim \mathcal{D}} \Big( \log \sum_{a'} \exp \Big( Q_\theta(s_i,\, a') \Big) - Q_\theta(s_i,\, a_i) \Big) \tag{30}$$

### D.4.2 AUXILIARY LOSS FOR CONSISTENCY IN THE WORLD MODEL

In order to avoid overburdening the latent states with extraneous information required to reconstruct the original input states, like in Model-based RL methods, we utilize self-supervised consistency loss functions as described in (Schwarzer et al., 2021; Ye et al., 2021). These functions aid in maintaining consistency within the transition and reward networks. For example, let us assume a state $s_i$ and the subsequent state $s_{i+1}$ resulting from action $a_i$. The latent state representations for the environment states $s_i$ and $s_{i+1}$ can be computed as $h_i$ and $h_{i+1}$ respectively. The latent state encoding $\hat{h}_{i+1}$ can be predicted using the transition module, $\hat{h}_{i+1} = \mathcal{T}_\theta(h_i,\, a_i)$. We minimize the squared error between the latent representation $h_{i+1}$ and $\hat{h}_{i+1}$, to ensure that the transition function $\mathcal{T}_\theta$ provides consistent predictions for the transitions in the latent space. In accordance with the approach detailed in (Ye et al., 2021), we use a separate encoding network, referred to as the target encoder, to compute target representations.

$$\mathcal{L}_{\mathcal{T}_\theta} = \mathbb{E}_{(s_i,\, a_i,\, s_{i+1}) \sim \mathcal{D}} \left[ \left( \hat{h}_{i+1} - h_{i+1} \right)^2 \right] \tag{31}$$

where

$$\hat{h}_{i+1} = \mathcal{T}_\theta(h_i,\, a_i)$$
$$h_i = \mathcal{E}_\theta(s_i)$$
$$h_{i+1} = \mathcal{E}_{\theta'}(s_{i+1})$$

The parameters of the target encoder, $\theta'$, are updated using an exponential moving average of the parameters of the base encoder, $\theta$, as follows.

$$\theta' \leftarrow \alpha\, \theta' + (1 - \alpha)\, \theta$$

Notably, we refrain from adding projection or prediction networks, as done in Schwarzer et al. (2021) and Ye et al. (2021), prior to calculating the squared difference as we did not observe any improvement by doing so.

Further, we also seek to minimize the mean squared error between the predicted reward $\mathcal{R}_\theta(h, a)$ and the actual reward observed $r$ in the training dataset $\mathcal{D}$.

$$\mathcal{L}_{\mathcal{R}_\theta} = \mathbb{E}_{(s_i,\, a_i,\, r_i) \sim \mathcal{D}} \left[ \left( \mathcal{R}_\theta(h_i,\, a_i) - r_i \right)^2 \right] \tag{32}$$

### D.4.3 FINAL LOSS FUNCTION

The final loss function to train D-TSN is a combination of Behavior Cloning loss and all the auxiliary losses defined above. It is given by:

$$\mathcal{L} = \mathbb{E}_\tau \Big[ \lambda_1 \mathcal{L}_Q + \lambda_2 \mathcal{L}_\mathcal{D} \Big] + \lambda_3 \mathcal{L}_{\mathcal{T}_\theta} + \lambda_4 \mathcal{L}_{\mathcal{R}_\theta} \tag{33}$$

where $\lambda_1, \lambda_2, \lambda_3$ and $\lambda_4$ serve as weighting hyperparameters.

### D.5 BASELINES

To evaluate the efficacy of Tree Search Network, we benchmark it against the following prominent baselines:

- **Model-free Q-network**: This allows us to assess the significance of integrating the inductive biases into the neural network architecture.
- **Model-based Search**: In this baseline, we assess the benefits derived from the joint optimization of the world model and the search algorithm by training the world models and value module independently of each other and utilizing them for online search.
- **TreeQN**: This comparison helps in highlighting the advantages of using a more advanced search algorithm that can execute a deeper search while maintaining similar computational constraints.

### D.6 IMPLEMENTATION DETAILS

In an effort to assess the distinctive elements of each method's design, we ensure uniformity in the number of parameters across all agents. This is achieved by integrating the submodules from D-TSN into the network architecture of each baseline. However, while the number of parameters are consistent, the way in which these submodules are utilized to construct the computation graph varies among the baselines. We provide their implementation details below:

### D.6.1 DIFFERENTIABLE TREE SEARCH NETWORK

D-TSN utilizes its submodules in alignment with the best-first search algorithm presented in Section 3.2. For our empirical evaluations, we set the maximum limit for search iterations at 10. Throughout the training process, the computation graph, formulated via online search, is optimized to accurately predict the Q-values. This optimization serves a dual purpose: it not only refines the Q-value predictions but also facilitates robust learning for the submodules when they are employed in context of online search. The loss function utilized for training is:

$$\mathcal{L}_{D-TSN} = \mathbb{E}_\tau \Big[ \lambda_1 \mathcal{L}_Q + \lambda_2 \mathcal{L}_\mathcal{D} \Big] + \lambda_3 \mathcal{L}_{\mathcal{T}_\theta} + \lambda_4 \mathcal{L}_{\mathcal{R}_\theta} \tag{34}$$

We compute the gradient for this loss as described in Equation (11).

### D.6.2 MODEL-FREE Q-NETWORK

In this baseline, the submodules are utilized to perform a one-step look-ahead search. The input state at the root node undergoes an expansion using the world model, and Q-values are computed using the Bellman equation represented as $Q(s, a) = Rew(h, a) + Val(h')$, where $h = Enc(s)$ and $h' = Tr(h, a)$. Intriguingly, this structure does encapsulate a basic inductive bias via the 1-step look-ahead search. However, in keeping with its model-free characteristic, auxiliary losses aren't employed for training the transition and reward model. The loss function for this model is:

$$\mathcal{L}_{QNet} = \lambda_1 \mathcal{L}_Q + \lambda_2 \mathcal{L}_\mathcal{D}$$

### D.6.3 MODEL-BASED SEARCH

For this approach, we employ the best-first algorithm showcased in Algorithm 1. However, there's a difference: the world model and the value module are trained independently, each focusing on their specific objectives. As outlined in Ye et al. (2021), we incorporate self-supervised consistency losses defined in Equations (31) and (32) as they improve the online search, even in cases where the world model is not jointly trained with the online search. The Q-values used for training are computed directly using the value module without performing online search during training. The loss function used for this approach is:

$$\mathcal{L}_{Search} = \lambda_1 \mathcal{L}_Q + \lambda_2 \mathcal{L}_\mathcal{D} + \lambda_3 \mathcal{L}_{\mathcal{T}_\theta} + \lambda_4 \mathcal{L}_{\mathcal{R}_\theta}$$

### D.6.4 TREEQN

In this baseline, the starting step is encoding the input state to its latent counterpart with the Encoder module. Following this, a full-tree expansion, based on a predefined depth $d$, is performed using both Transition and Reward modules. The values at the leaf nodes are then backpropagated to the root node via the Bellman equation, as discussed in the Backup phase in Section 3.2. The root node Q-values serve as the final output, that is utilized for training. Given the exponential growth of TreeQN's computation graph with an increase in depth $d$, we choose a depth of 2 for both Procgen and navigation domains in our analysis, as used in TreeQN's original code base (Farquhar et al., 2018). Notably, greater depths, such as 3 or more, are infeasible since the resulting computation graph exceeds the memory capacity (roughly 11GB) of a typical consumer-grade GPU. The associated loss function, as adapted from the original paper, is:

$$\mathcal{L}_{TreeQN} = \lambda_1 \mathcal{L}_Q + \lambda_2 \mathcal{L}_\mathcal{D} + \lambda_3 \mathcal{L}_{\mathcal{R}_\theta}$$

It is important to note that every method is trained with same datasets using their respective loss functions. We fine-tune the hyperparameters, $\lambda_1, \lambda_2, \lambda_3$ and $\lambda_4$, using grid search on a log scale.

### D.7 ISSUE WITH BASELINE NORMALIZED SCORE

Within the context of Atari games, the Baseline Normalized Score (BNS) is frequently utilized to evaluate the performance of agents. When human players are used as the baseline, it is often termed Human Normalized Score. The primary allure of BNS lies in its capacity to offer a relative assessment of an agent's performance, comparing it against a standard benchmark—this could be human players or even another agent.

One of the primary benefits of the BNS is its ability to provide a consistent metric across different games, addressing the difference in scale inherent in raw scores. By enabling the calculation of the average BNS across multiple games, we gain insight into the overall efficacy of an agent. This not only facilitates direct performance comparisons between diverse agents and methodologies but also paints a picture of how the agent's abilities stack up against human standards.

To derive the BNS, we start by logging the agent's raw score in an Atari game. This raw score is then normalized against a baseline score, derived from baseline agent's performance on the same game. By dividing the agent's score by the baseline's score (and sometimes subtracting the score of a random agent), we get a relative metric. Mathematically, this can be represented as:

$$BNS(\pi) = \frac{S_\pi - S_R}{S_B - S_R} \tag{35}$$

Here, $S_\pi, S_B$ and $S_R$ denote the raw scores of the agent, the baseline policy, and a random policy, respectively. Interpretation-wise, a BNS of 1 indicates parity with the baseline. Values exceeding 1 signify that the method outperforms, while those below 1 indicate that the method underperforms relative to the baseline.

Nevertheless, the BNS has its frailties. It inherently presumes the baseline policy will always surpass the performance of the random policy. But there can be instances, contingent on the environment or the specific baseline policy, where this isn't the case. In scenarios where the baseline policy underperforms the random policy, the BNS results in a negative denominator. This poses a predicament: even if our agent's policy performs better than the random policy, the BNS unfairly penalizes it. In our experiments with Procgen, we observed that for 2 out of the 16 games, namely heist and maze, the baseline policy underperformed compared to the random policy. Given these pitfalls, our evaluations pivot towards a more robust metric: the Z-score. The Z-score, often termed as the "standard score", provides a statistical measurement that describes a value's relationship to the mean of a group of values. It is measured in terms of standard deviations from the mean. If a Z-score is 0, it indicates that the data point's score is identical to the mean score. Z-scores may be positive or negative, with a positive value indicating the score is above the mean and a negative score indicating it is below the mean.

