# OpenReview forum: "Learning to Search from Demonstration Sequences"
_ICLR.cc/2025/Conference — ICLR 2025 Oral_

### Official Review · Reviewer_iWz4 · 2024-11-01

**Soundness:** 3
**Presentation:** 2
**Contribution:** 2
**Rating:** 6
**Confidence:** 3

**Summary:**

This study presents a new differentiable tree search network comprising multiple submodules to estimate Q values. The network is optimized by minimizing the distance between the estimated Q values and the ground-truth Q values within a provided dataset. To address the substantial changes in the tree structure resulting from updates to the Q value function, a stochastic expansion policy is introduced to guide expansions in the search process. This policy ensures the continuity of the parameter space irrespective of changes in the tree structure.

**Strengths:**

i) This work introduces the integration of the algorithmic inductive bias from the best-first search algorithm into a differentiable network, enabling automatic guidance of the tree search process through gradient backpropagation.

ii) This work underscores the significance of maintaining continuity of both the parameter space and the loss function which are dependent on the tree structure. To address this, the authors advocate for the adoption of a stochastic expansion policy to fulfill these prerequisites.

iii) The experiment results are compelling. And it is particularly noteworthy to see the success achieved in tasks involving LLM.

**Weaknesses:**

i) Previous methods have introduced diverse differential network architectures to integrate different search algorithms into networks, as mentioned in the related works. It is unsurprising that integrating the best-first algorithm into the network has also yielded success. Thus, it would be beneficial to compare the architectural variances between this method and previous methodologies.


ii) This work trains the overall network using an offline dataset. However, as extensively deliberated in preceding offline RL studies, this paradigm may get stuck when facing out-of-distribution states or actions. Thus, a comparative analysis between online training and offline training for the newly proposed network architecture could provide valuable insights.

**Questions:**

Do differences in the convergence rates of submodules exist? If present, could these differences impact the overall network's performance?

---

> ### Author Response · Authors · 2024-11-18
>
> Dear Reviwer iWz4,
>
> We express our gratitude for your thorough review and insightful comments on our paper. We are glad that you find our work **interesting** and the **experimental results compelling**. We address the concerns you raised as follows:
>
> > **W1**: Previous methods have introduced diverse differential network architectures to integrate different search algorithms into networks, as mentioned in the related works. It is unsurprising that integrating the best-first algorithm into the network has also yielded success.
>
> We thank the reviewer for raising this question. There is a key distinction between the architectures mentioned in related works and our proposed method – they use **a separately learned world model**, while our method jointly learns a world model with the tree search. Moreover, our method constructs a search tree dynamically, instead of a fixed structure as in TreeQN, which allows it to search deeper compared to a shallow full tree, given the same computation budget.
>
> In summary, we are the first work that proposes to differentiably construct a dynamic search tree and learn a world model jointly with the search algorithm. Further, we formulate the tree expansion as another decision-making task and solved it using REINFORCE. These unique properties set our work apart from existing related works.
>
>
> > **W2**: Discussion of Online RL
>
> Thank you for raising this question. Our work primarily focuses on Offline RL settings and aims to learn to search from demonstration sequences only. The proposed method can be applied to an online setting without additional modifications. Although, in an online setting, the benefit of jointly learning a world model might be less significant, as the agent can explore unseen regions, collect data, and learn the policy. However, the proposed method would still benefit from its strong search inductive bias and improve the sample efficiency for training.
>
> > **Q1**: Do differences in the convergence rates of submodules exist? If present, could these differences impact the overall network's performance?
>
> We do observe different convergence rates for different submodules. On the navigation problem, we observe that the reward function quickly converges after around 10k optimization steps, while the transition function and value function converge after around 40k steps. The difference in convergence rate may related to the difficulty of learning the submodule, for example, the reward function in the navigation problem is easy to learn as most rewards are 0 and only terminal states may have a positive reward. Generally, we observe that the performance of the overall network continues to increase even if some submodules have not converged yet.
>
> We deeply value the thoughtful and constructive feedback you have provided. We sincerely hope we have addressed your concerns and raised your impression of our work. We are happy to clarify any other questions if required.

---

> > ### Comment · Reviewer_iWz4 · 2024-11-19
> >
> > I thank the author for the responses. I acknowledge that these replies have solved my concerns. I urge the author to highlight the architecture difference with the previous work (i.e. TreeQN) in the revised version. And I also recommend the author to include the discussion about the convergence for each submodule in the experiment part. In light of this, I am inclined to raise the score from 5 to 6.

---

> > > ### Author Response · Authors · 2024-11-20
> > >
> > > Dear reviewer iWz4,
> > >
> > > Thank you again for spending time reviewing our paper and engaging in the rebuttal discussion. We are pleased to know that you found our response useful. We will include a discussion on the architectural differences between D-TSN and other models, as well as an analysis of the convergence rates of the different submodules in the rebuttal revision.

---

> > ### Comment · Reviewer_KW9a · 2024-11-23
> > **Re: Response to questions and suggestions**
> >
> > Thank you for taking the time to respond to my questions. I have updated my score of the paper. Good luck!

---

### Official Review · Reviewer_KW9a · 2024-11-03

**Soundness:** 3
**Presentation:** 3
**Contribution:** 4
**Rating:** 8
**Confidence:** 4

**Summary:**

This paper introduces a differentiable tree search and planning network to alleviate the suboptimal search results that arise when using only action and observation sequences to learn to plan in many modern data-driven planning problems. The tree search basically synergizes submodules in the form of an encoder, value, reward and transition functions from a network by inducing the bias of a best first search into the neural architecture.

Arguing that for a slight perturbation in network parameters, the implementation of the loss function in equation (2) could generate a tree structure that causes the loss function to become noisy, the authors equated this to a lack of continuity in the loss function space. I think they mistook stochasticity in gradient propagation with discontinuity. The whole premise of the contribution of the paper is based on this assumption that is barely proven to be true or false before the authors dived into a host of mathematical analysis that resulted in the loss function on Line 272 (please number all your equations to allow for easy referencing in the future). As a result of this key oversight, it is not clear to this reviewer that the whole contribution of the paper is warranted.

As a result, I am assigning a score of fair to this contribution until I see a well-laid out argument for why this new invention is necessary.

**Strengths:**

+ This paper introduces a differentiable treee seach and planning network to alleviate the suboptimal search results that arise when using only action and observation sequences to learn to plan in many modern data-driven plannning problems. The tree search basically synergizes submodules in the form of an encoder, value, reward and transition functions from a network by inducing the bias of a best first search into the neural architecture.

+ I love the motivation stated for constructing an search tree expansion policy that is stochastic in nature. But the justification for why it ensures the continuity of the loss function when the search tree is not fixed is missing. I am referring to lines 55-56.

+ I love the conceptualization and the synergy of REINFORCE, mathematical mechanisms to reduce variance in the REINFORCE Loss owing to possibly biased estimates, the continuity proof (though I have questions hanging over the proof of Lemma B.3 to be fully satisfied with this poposition) .

+ I love that the conclusion section meticulously summarizes the problem, contributions, and shortcomings. Kudos to the authors.

**Weaknesses:**

While I do love the mathematical contributions of the paper, I think there are essentials that are missing in the logic, organization, and flow of arguments that require a thorough review before this paper makes it into an acceptance. A principal one is the following (mentioned in the summary box but repeated here. The authors would do well to alleviate my concerns): Arguing that for a slight perturbation in network parameters, the implementation of the loss function in equation (2) could generate a tree structure that causes the loss function to become noisy, the authors equated this to a lack of continuity in the loss function space. I think they mistook stochasticity in gradient propagation with discontinuity. The whole premise of the contribution of the paper is based on this assumption that is barely proven to be true or false before the authors dived into a host of mathematical analysis that resulted in the loss function on Line 272 (please number all your equations to allow for easy referencing in the future).

+ The claim in the last paragraph of Theorem 3.1 that slight changes in the network parameters could cause discontinuity in the tree structure seems anecdotal and not backed up by a solid proof. I would love to see a concrete reasoning (analytical proof or abundant empirical  proofs) behind this claim that warrants section 3.5 and Appendix C.

+ Grammatical errors fly out of the page hither and yon throughout the paper; the uthors would do well to carefully organize their arguments, present their logic convincingly throughout the paper, and punctuate and label every equation appropriately!

+ The logic in the paper could use a more thoughful presentation. Here is an example critique:
     - In the "introduction", it is stated in the first paragraph that constructing a search tree from expert data is infeasible due to lack of practicality or scarcity. The authors make an assumption that a search tree is a principal prerequisite for information retrieval (IR) without any justification as to why it may be better than alternative IR methods. Then in the second paragraph, they mentioned how search and planning could be better executed in the presence of a simulator or world mode. While I find this premise alluring, I find it disingenious that the authors claimed that the search could be incomplete because the search process may visit regions unexplored during training. I think the reasoning here is incomplete and should be revisited by the authors.

**Questions:**

+ How is the full expansion mechanism in this work not a shallow tree (in comparison to the TreeQN) that you mentioned in the RELATED WORKs section since you mentioned that you have a fixed depth in your search protocol?

+ When the encoder, value, reward, and transition functions are unavailable, how do they get jointly optimized with the search algorithms?

+ In paragraph III, the justification for why end-to-end learning improves the reliability of world-models used in preconditioning the search process is not given. Please address this.

+ Lines 124-126: "Dividing the network into these submodules reduces
total learnable parameters and injects a strong search inductive bias into the network architecture, preventing overfitting to an arbitrary function that may align with the limited available training data." Where is the proof for this?

+ Is there no way the illustration in Figure 2 can be moved to the main text from the Appendix? Probably as a subfigure on page one or so would be really nice! Also, I think the algorithm should justifiably be compressed (e.g. as a pseudocode or flow chart) somewhere in the main text. An elaborate version can be embedded in the Appendix if space is an issue.


+ What metric are you using to report the measured quantities in Table 1? Can you include it in the heading/sub-headings?


+ Appendix B, Lemma B.3: I'm sorry, is there supposed to be a running sum over all of x in the equation you wrote?

+ Line 915, I think you meant Lemma B.3, not Theorem B.3, no?

---

> ### Author Response · Authors · 2024-11-18
> **Response Part 1**
>
> Dear Reviewer KW9a,
>
> We thank you for spending time thoroughly reviewing our paper and providing constructive feedback, which helped us improve our work. We are pleased to know that you loved the **motivation** and **overall idea** behind our work. We address the concerns you raised as follows:
>
> > **W1.** Proof or empirical evidence on “slight changes in the network parameters could cause discontinuity in the tree structure”
>
> We have provided the proof that 'the continuity of the Q-function is not guaranteed when the tree structure is not fixed' by giving a counter-example in Appendix B of the revised manuscript of the paper and have updated it in this submission. Please have a look. We are happy to provide further clarifications if needed.
>
> > **W2**: Fix grammatical errors and label every equation.
>
> We thank you for pointing out these issues. We have performed a thorough grammar check and have labeled every equation in the revised manuscript.
>
> > **W3.1**: Why a search tree is a principal prerequisite for information retrieval?
>
> The use of a search tree is adopted in various works, such as [1, 2, 3], in decision-making and planning problems. Search trees also show improvements in *question answering-type information retrieval* tasks, such as  [4].  We are happy to discuss other 'IR methods' with the reviewer and will include them in the revised manuscript with specific references.
>
> > **W3.2**: I find it disingenious that the authors claimed that the search could be incomplete because the search process may visit regions unexplored during training. I think the reasoning here is incomplete and should be revisited by the authors.
>
> Thank you for pointing out the issue of reasoning flow in paragraph 1-3. In paragraph 2, we stated that 'the *separately* learned **world models** are often incomplete' – by “incomplete” we mean that the world model is not trained on the complete state space and may perform badly on the states not seen during the training phase. Consequently, by using such world models, the search would suffer from the issue of compounding errors, particularly when the search is expanded into these unseen regions. In the next paragraph, we mentioned that end-to-end learning can improve the reliability of the world model. However, we might have not provided the clear reasoning in the previous manuscript. We provide the intended reasoning here: the advantage of end-to-end learning is that the world model and the search algorithm are trained together with a loss function that improves final performance of the resulting policy. In doing so, the search algorithm would learn to **avoid going into state spaces where the world model performs badly** (since going into these states would result in low final performance). We have added the above reasoning in the revised manuscript.
>
>
> > **Q1**: How is the full expansion mechanism in this work not a shallow tree ... since you mentioned that you have a fixed depth in your search protocol?
>
> We believe there is a misunderstanding: we **do not** perform full expansion in D-TSN and we **do not** have a fixed depth. Instead, we perform best-first search to construct the tree for a fixed number of tree nodes (or search iterations), where the resulting tree structure can be deep and sparse as compared to a fully expanded shallow tree (as done in TreeQN).
>
> > **Q2**: When the encoder, value, reward, and transition functions are unavailable, how do they get jointly optimized with the search algorithms?
>
> The encoder, value, reward, and transition functions are invoked by the search algorithm to finally derive the Q-values of the root node. Correspondingly, we use REINFORCE to train these submodules to minimize the expected loss on the Q-values. Please refer to Equation 16 in the revised manuscript for the loss function used by D-TSN. Also, please refer to Appendix D for a more visual-friendly illustration of the pipeline.
>
> > **Q3**: Why end-to-end learning improves the reliability of world models used in search?
>
> Please see our response to W3.2.

---

> > ### Comment · Reviewer_KW9a · 2024-11-18
> > **Re: Response Part 1**
> >
> > (I will add my response to W1 later after I shall have carefully read it).
> >
> > W3.2 Looking forward to reading this in your revision. It is not there in the current attachment: https://openreview.net/notes/edits/attachment?id=DyBgbT0jV9&name=pdf
> >
> > With respect to your response to Q1, please do ensure that the paper reflects this explanation that you have just provided.

---

> > > ### Author Response · Authors · 2024-11-19
> > >
> > > Thank you for your prompt reply and the time you've spent reviewing our response and revision. Your effort is greatly appreciated.
> > >
> > > We have updated a new revision to further address W3.2 in paragraph 2, our response to W3.2 in the rebuttal is reflected in line 42-50 in the revision:
> > >
> > > > line 42: However, when separately learned from demonstration sequences, such world models often suffer from compounding errors, and cannot cover the complete state space. Using such world models for search can be unreliable, particularly if the search expands into parts of the space that are not seen during training.
> > >
> > > > line 47: One way to increase the reliability of the world model is to perform end-to-end learning, where the world model is trained with the search algorithm. The advantage of end-to-end training is that the search algorithm will learn to avoid going into states where the world model is inaccurate since this would result in lower performance.
> > >
> > > Thank you for asking Q1 to help us clarify the details in D-TSN, our response to Q1 is reflected in the paper:
> > >
> > > "fixed depth" is mentioned twice in the manuscript in line 90 and line 193, which are referring to TreeQN
> > >
> > > > line 90: Another notable work, **TreeQN**, incorporates search inductive bias into the network by fully expanding a search tree to a **fixed depth**...
> > >
> > > > line 193: The theorem applies to **TreeQN**, which performs a full tree expansion to a **fixed depth**...
> > >
> > > In line 129 we explained how we construct a search tree in D-TSN:
> > >
> > > > During expansion, the search tree expands iteratively for a set number of expansion steps, **where each step expands a node in the search tree**.
> > >
> > > In the experiment sections, we state the number of search iterations by `n_itr` and state in the table heading, e.g.
> > >
> > > > Table 1 ... D-TSN is trained and evaluated with 8 search iterations ...
> > >
> > > > line 387: For our evaluations, we perform 10 search iterations for each input state

---

> ### Author Response · Authors · 2024-11-18
> **Response Part 2**
>
> > **Q4.** Proof for "Dividing the network into these submodules reduces total learnable parameters and injects a strong search inductive bias into the network architecture, preventing overfitting to an arbitrary function that may align with the limited available training data."?
>
> Firstly, in D-TSN, we construct a search tree by reusing submodules including encoder, value, transition, and reward function, this results in a smaller number of total network parameters as compared to constructing a tree of the same size without reusing the submodules.
>
> In a discretized parameter space, we can apply Occam's Razor to derive a generalization bound proportional to the number of parameters. According to Occam's Razor principles, simpler models (those with fewer parameters) are favored, as they tend to generalize better to unseen data [1].
>
> For continuous parameter space, we provide a proof sketch based on VC-dimension [5]. Consider a complex hypothesis space $H_{complex}$, a model from this space is flexible enough to fit any arbitrary functions, including those that align with the noise in training data. By dividing the network into submodules and reusing them multiple times in the search algorithm, we construct a constrained hypothesis space $H_{bias}$. The restriction of this bias leads to $VC(H_{bias}) < VC(H_{complex})$, which means $H_{bias}$ has a lower degree of freedom to fit arbitrary complex functions. Given $H_{complex}$ and $H_{bias}$, Occam’s Razor would favor $H_{bias}$ for its simplicity, as simpler models (those with lower VC dimension) tend to generalize better to unseen data. Thus, by injecting inductive search bias into the model design, we effectively reduce the risk of overfitting due to the decreased VC dimension and alignment with Occam's Razor principles.
>
> > **Q5**: Flow chart of the algorithm in the main text.
>
> Thank you for this suggestion, we have added an illustration of the algorithm in the main text of the revised manuscript.
>
> > **Q6**: What metric is used in Table 1?
>
> Quantities reported in Table 1 are success rate, i.e. whether the generated expression is valid, and results in the game of 24. We have updated the Table heading in the revised manuscript.
>
> > **Q7**: Appendix B, Lemma B.3: I'm sorry, is there supposed to be a running sum over all of x in the equation you wrote?
>
> No, by Lemma B.1, B.2 and B.3, we show that h(x)=max(f(x), g(x)) is continuous, which is the building block of the Q-function represented by TreeQN – the backup phase calculates the value of each node by calculating the maximum Q-values of its children, which resembles h(x)=max(f(x), g(x)). We are happy to clarify any further questions.
>
> > **Q8**: Typos
>
> Thank you for pointing it out. We have fixed these in the revised manuscript.
>
>
> We sincerely appreciate the constructive critique you have provided and hope that our responses and the modifications to the manuscript adequately address your insightful feedback and increases your impression and confidence in our work. We are happy to provide any further clarifications if needed.
>
> [1] Silver, David, et al. "Mastering the game of Go with deep neural networks and tree search." nature 529.7587 (2016): 484-489.
>
> [2] Mohanan, M. G., and Ambuja Salgoankar. "A survey of robotic motion planning in dynamic environments." Robotics and Autonomous Systems 100 (2018): 171-185.
>
> [3] Duarte, Fernando Fradique, et al. "A survey of planning and learning in games." Applied Sciences 10.13 (2020): 4529.
>
> [4] Yao, Shunyu, et al. "Tree of thoughts: Deliberate problem solving with large language models." Advances in Neural Information Processing Systems 36 (2024).
>
> [5] Blumer, Anselm, et al. "Learnability and the Vapnik-Chervonenkis dimension." Journal of the ACM (JACM) 36.4 (1989): 929-965.

---

> ### Author Response · Authors · 2024-11-24
>
> Dear Reviewer KW9a,
>
> We thank you for your thorough review of our paper and for providing constructive feedback that has significantly contributed to its improvement. Your insights have been invaluable in helping us refine our work.
>
> We sincerely hope that our responses have sufficiently addressed the issues you highlighted in your review and follow-up comments. As the author-reviewer discussion period approaches its end, please do not hesitate to let us know if there is anything further we could do to improve your impression and final rating of our work.

---

### Official Review · Reviewer_7dAQ · 2024-11-04

**Soundness:** 4
**Presentation:** 2
**Contribution:** 3
**Rating:** 8
**Confidence:** 4

**Summary:**

This paper presents a novel differentiable neural tree search architecture that can be learned directly from trajectories of data in a supervised manner. It essentially introduces a search-like inductive bias within the weights of the neural network. The proposed algorithm builds on TreeQN (Farquhar et al., 2018), and is crucially different since the proposed method allows building a tree structure that can stochastically sample from the action space, and not just be a fixed tree structure as in TreeQN. The authors empirically evaluate the strength of their approach by comparing against a Model-free Q network, a Model-based search method that trains the individual modules separately, and TreeQN, and report performance gains in various RL environments.

**Strengths:**

- Builds on TreeQN and improves a significant limitation of the prior work, i.e., having a fixed tree structure. In reality, search algorithms should attempt to filter large action spaces and focus computation on promising variations in the tree. The proposed work gets around this limitation by sampling from the action space, and using REINFORCE to differentiate through the discontinuity of sampling.
- Strong empirical evidence that the proposed method improves on TreeQN, and having the modules trained separately.
- Strong ablation results showing the effectiveness of the proposed method.

**Weaknesses:**

- Seeing how the approach handles large action spaces remains an empirical question, since currently, there is no policy that directly outputs a distribution over actions, instead the method requires the application of the transition network and the reward network for every action, which is not scalable to settings like, say, Go.
- The proposed method is mostly applicable to discrete action spaces with deterministic environments. Improving on this remains a future empirical question.

**Questions:**

- Why did the authors choose to use REINFORCE? Did the authors experiment with using methods like Gumbel-Softmax for sampling?
- How are the compute budgets between model-based search, model-free Q network and D-TSN equalized for a fair comparison?
- In Line 22, the authors say they “introduce an effective variance reduction technique”. Isn’t this just Guez et al. (2018)? The way this is written in some places suggests that this is novel.
- Line 45,46 seems contradictory with lines 48,49. If learning from demonstration sequences fails due to compounding errors due to the agent getting into states not during training, then the training distribution of the proposed method is also not sufficiently covered by the training distribution. I understand the CQL term attempts to address this issue.
- What are the scores of the PPG sub-optimal policy?
- Table 3 reports Mean Scores and Mean Z-Scores, but no standard deviations or error bars?

---

> ### Author Response · Authors · 2024-11-18
> **Response Part 1**
>
> Dear Reviewer 7dAQ,
>
> We thank you for your insightful and constructive feedback on our manuscript. We appreciate your recognition of the novelty and practical application of our approach in addressing the complexity issue in TreeQN and its empirical evaluation. Below, we address your concerns to clarify and improve our work.
>
> > **W1**: How the approach handles large action spaces remains an empirical question
>
> We agree with the reviewer that D-TSN can be expensive when the action space is large, as we need to evaluate every action’s Q-value. The computational cost grows linearly as the action space grows, this is a general challenge for value-based reinforcement learning methods when dealing with large action spaces. One potential way to adapt to a large action space is to have a Q-network that outputs the Q-values of all actions in one forward pass. Adapting D-TSN to use a Q-network to evaluate action branches is straightforward, and the search algorithm will remain largely the same.
>
> > **W2**: Application beyond discrete action space and deterministic environments uncertain.
>
> We agree that the current implementation of D-TSN is not aimed at stochastic environments and continuous action spaces. However, we briefly discuss how the current work might adapt to such scenarios below:
> - For stochastic environments, we could take inspiration from existing works like [1], where we can incorporate an intermediate 'afterstate' in the search tree. In this approach, a state first deterministically transits to an intermediate 'afterstate', and then branches stochastically to the next state, which allows it to address the stochasticity in the environment.
> - For continuous action spaces, search and planning methods often do not fit well into these problems. A popular approach is to discretize the action space so that it can fit into the search and planning framework. We could also perform multi-level discretization to trade-off between the depth and width of the search tree. For example, if a single action is defined by $k$ action variables $(a_1, .., a_k)$, we may discretize and branch on $a_1$ first, followed by discretization and branching on $a_2, ...,$ all the way to $a_k$, converting each action into a $k$-level action, and correspondingly increasing the depth of the tree. Our best-first search action selection will still result in a sparse tree for this scheme.
>
> > **Q1**: Why choose REINFORCE, what about gumbel-softmax for differentiable sampling?
>
> In the paper, we formulate the tree expansion as another decision-making task, and REINFORCE comes as a natural solution for such tasks. We agree that using the gumbel-softmax trick for differentiable sampling in Eq. 7 (in revised manuscript) can be a possible alternative. However, gumbel-softmax may require additional tuning of the temperature and dynamically adjusting it during training, which may make training more difficult. Nonetheless, it is an interesting approach that we can explore in the future.
>
> > **Q2.** Computation budgets of different methods?
>
> We report the average time taken per training step and the total training time for a sample Procgen environment on a RTX 2080Ti GPU in the following table:
> | Solver| Time Taken Per Training Step | Total Training Time |
> |----------------------|------------------------------|---------------------|
> | Model-free Q-network | 43ms| 4h0m|
> | TreeQN (depth=2)| 357ms| 23h45m|
> | D-TSN (n_itr=5)| 153ms| 9h30m|
> | D-TSN (n_itr=10)| 294ms| 17h30m|
>
> > **Q3.** Writing suggestion.
>
> Thank you for pointing it out. We have modified the sentence in the revised manuscript to _“To construct the search tree, we employ a stochastic tree expansion policy and formulate it as another decision-making task. Then, we optimize the tree expansion policy via REINFORCE with an effective variance reduction technique for the gradient computation.”_
>
> > **Q4**: If learning from demonstration sequences fails due to compounding errors due to the agent getting into states not during training, then the training distribution of the proposed method is also not sufficiently covered by the training distribution.
>
> We agree with the reviewer that our method cannot cover more training distribution as compared to training the world model separately. The advantage of joint training is that the world model and the search algorithm are trained together with a loss function that optimizes the final performance. In doing so, the search algorithm would learn to avoid going into state spaces where the world model performs badly (since going into these states would result in low final performance). When the world model and the search algorithm are learned separately, the search would not avoid those unseen state spaces and thus may harm the performance.

---

> ### Author Response · Authors · 2024-11-18
> **Response Part 2**
>
> > **Q5**: What are the scores of the PPG sub-optimal policy?
>
> The performance of the policy we used to collect trajectories is reported in the table below:
> | Game| Demonstration Policy |
> |------------|-----------------------|
> |Bigfish|26.98|
> |Bossfight|11.13|
> |Caveflyer|9.66|
> |Chaser|8.11|
> |Climber| 9.95|
> | Coinrun| 8.40|
> | Dodgeball| 8.88|
> | Fruitbot| 23.52|
> | Heist| 3.49|
> | Jumper|6.54|
> | Leaper|8.92|
> | Maze|7.00|
> | Miner|10.84|
> | Ninja| 9.41|
> | Plunder| 17.67|
> | Starpilot| 22.33|
>
> > **Q6.** Std and error bars of Table 3?
>
> We are in process of rerunning the experiments to calculate the std and error bars and will update once finished.
>
> We sincerely appreciate the constructive critique you have provided and hope that our responses and the modifications to the manuscript adequately address your insightful feedback and increases your impression and confidence in our work.
>
> [1] Antonoglou, Ioannis et al. “Planning in Stochastic Environments with a Learned Model.” International Conference on Learning Representations (2022).

---

> > ### Comment · Reviewer_7dAQ · 2024-11-18
> >
> > Thank you for such detailed and thorough responses to my questions and fixing some of the clarity and writing issues.
> >
> > > In doing so, the search algorithm would learn to avoid going into state spaces where the world model performs badly ...
> >
> > This makes a lot more sense, I appreciate the revision.
> >
> > > Figure 1
> >
> > This is a great addition to the paper, and makes it significantly easier to follow.
> >
> > > We report the average time taken per training step and the total training time for a sample Procgen environment on a RTX 2080Ti GPU in the following table
> >
> > **Q7.** Given the numbers in the table, would the authors think the performance difference between `Model-free Q-network` and `D-TSN` could be simply explained by the amount of computation?
> >
> > > We are in process of rerunning the experiments to calculate the std and error bars and will update once finished.
> >
> > Thank you! I understand that getting new numbers quickly adds work, and I appreciate adding these metrics.
> >
> > ---
> >
> > Pending an adequate answer to **Q7** and the addition error bars, I would be inclined to raise my score since the authors have adequately answered my other questions and addressed weaknesses.

---

> ### Author Response · Authors · 2024-11-19
>
> Thank you for your prompt reply. We are pleased to hear that you find our response and revisions have added clarity to the manuscript.
>
> > Given the numbers in the table, would the authors think the performance difference between Model-free Q-network and D-TSN could be simply explained by the amount of computation?
>
> This is a great question. We do not think the performance difference is merely due to computation. First, D-TSN is a search-based algorithm, which allows it to spend more inference-time computation by constructing a larger search tree. Model-free Q-network however, cannot scale at inference time as it simply gives the Q-values in one forward pass without constructing any structures that we can scale up. Model-free Q-network can be scaled up by increasing the model size and adding more training data, which requires more data collection and is harder to implement than inference-time scaling.
>
> Compared with other search-based methods such as TreeQN, D-TSN spends an equal or less amount of computation while achieving better performance, which demonstrates the efficiency of D-TSN's search mechanism.

---

> > ### Comment · Reviewer_7dAQ · 2024-11-19
> >
> > > which requires more data collection and is harder to implement than inference-time scaling.
> >
> > Thank you for answering my question, this does seem like a key strength of D-TSN.
> >
> > I will wait for the numbers in Table 3, please let me know when the revision comes.

---

> > > ### Author Response · Authors · 2024-11-20
> > >
> > > Dear Reviewer 7dAQ,
> > >
> > > We have rerun evaluations for Model-free Q-network, Model-based Search, TreeQN, and D-TSN on Procgen, 1000 runs per game, and updated Table 3 with standard errors in the rebuttal revision. Thank you again for spending time thoroughly reviewing our paper.

---

> > > > ### Comment · Reviewer_7dAQ · 2024-11-20
> > > >
> > > > Thank you for your engagement and discussions. The author's have adequately answered my questions and addressed most weaknesses, and for that reason I'm inclined to raise my score from a 6 to an 8.

---

### Official Review · Reviewer_tL81 · 2024-11-04

**Soundness:** 4
**Presentation:** 4
**Contribution:** 3
**Rating:** 10
**Confidence:** 4

**Summary:**

This paper introduces the Differentiable Tree Search Network (D-TSN), a novel neural network architecture designed to learn search strategies from sequences of demonstrations without access to explicit search trees. D-TSN integrates the inductive bias of a best-first search algorithm into its structure, enabling the joint learning of essential planning submodules, including an encoder, value function, and world model. To construct the search tree, the authors employ a stochastic tree expansion policy, formulating it as a decision-making task optimized via the REINFORCE algorithm. They introduce a variance reduction technique using a telescoping sum to address high variance in gradient estimates. D-TSN is applicable in scenarios where the world model is known or needs to be jointly learned with a latent state space. The authors evaluate D-TSN on tasks from both scenarios, including the Game of 24, a 2D grid navigation task, and Procgen games. Experimental results demonstrate that D-TSN is effective, particularly when the world model with a latent state space is jointly learned, outperforming baselines in terms of success rate and generalization capabilities.

**Strengths:**

1. Novel Architecture: The paper proposes a novel neural network architecture, D-TSN, which embeds the inductive bias of a best-first search algorithm, allowing for the end-to-end learning of planning components from demonstration sequences.

2. Joint Learning of Planning Components: D-TSN jointly learns the encoder, value function, and world model. This is advantageous when the world model is not given but is needed to be learned from data.

3. Variance Reduction Technique: Authors use an effective variance reduction technique using a telescoping sum in the REINFORCE algorithm to addresses the high variance associated with policy gradient methods.

4. Comprehensive Experiments: The method is applied to a wide variety of tasks, such as reasoning problems, navigation, and game environments, supporting the claim that it is versatile and effective across domains.

5. Improved Performance: The authors show that D-TSN outperforms baselines, showing its problem solving performance in challenging tasks with limited supervision, especially in jointly learned world model settings.

**Weaknesses:**

1. Limited to Deterministic Environments: The current implementation is restricted to deterministic decision-making problems with discrete action spaces.

2. Computational Complexity: The computational complexity for the approach  might be high, because it consists of constructing search trees and performing REINFORCE updates. This can be a problem especially when applying for deeper trees or larger action spaces.

3. Scalability: scalability is not thoroughly analyzed for longe-horizon tasks or higher dimensional state space.

**Questions:**

1. How would you extend D-TSNs to use for solving stochastic decision making problems? What modification would be required in the current work to accommodate stochastic transitions and rewards?

2. Could you provide more details on how computationally expensive D-TSN is and how does the method scale w.r.t. the action space size and the search tree depth?

---

> ### Author Response · Authors · 2024-11-18
>
> Dear Reviewer tL81,
>
> Firstly, we express our gratitude for your thorough review and insightful comments on our paper. Your recognition of the **novelty** and **contribution** of our work is greatly appreciated. We have taken your feedback as an opportunity to further refine our manuscript. We address your concerns below:
>
> > **W1**: "Limited to deterministic Environments"
>
> We agree that the current implementation of D-TSN is not aimed at stochastic environments and continuous action spaces. However, we briefly discuss how the current work might adapt to such scenarios below:
> - For stochastic environments, we could take inspiration from existing works like [1], where we can incorporate an intermediate 'afterstate' in the search tree. In this approach, a state first deterministically transits to an intermediate 'afterstate', and then branches stochastically to the next state, which allows it to address the stochasticity in the environment.
> - For continuous action spaces, search and planning methods often do not fit well into these problems. A popular approach is to discretize the action space so that it can fit into the search and planning framework. We could also perform multi-level discretization to trade-off between the depth and width of the search tree. For example, if a single action is defined by $k$ action variables $(a_1, .., a_k)$, we may discretize and branch on $a_1$ first, followed by discretization and branching on $a_2, ...,$ all the way to $a_k$, converting each action into a $k$-level action, and correspondingly increasing the depth of the tree. Our best-first search action selection will still result in a sparse tree for this scheme.
>
>
> > **W2**: "The computational complexity for the approach might be high".
>
> We agree that constructing search trees is a computationally expensive operation, this is the nature of search-based approaches. However, search-based methods come with added performance improvements over model-free methods which is evident from our wide range of experiments. Notably, our method is *more computationally efficient than other search-based methods* such as TreeQN as we construct the tree by best-first search instead of full tree expansion. Moreover, in D-TSN, we can tune the performance-computation trade-off (by changing the number of expansion steps) depending on the computation requirements of the problem.
>
> > **W3**: "scalability is not thoroughly analyzed for longe-horizon tasks or higher dimensional state space"
>
> Long-horizon tasks pose challenges such as delayed rewards, which may require doing more search iterations to look ahead further. This is a general challenge for reinforcement learning methods. Notably, the computational cost of our method *scales linearly as the number of search iterations grows* and offers a computationally-friendly approach to scaling to long-horizon problems compared to TreeQN (which scales exponentially in tree depth).
>
> On the other hand, high dimensional states may require larger networks for each submodule in our framework. In terms of overall computational cost, the D-TSN framework grows by a constant factor as submodules become larger.
>
> > **Q1.** How to extend to stochastic decision-making problems?
>
> Please see our response to **W1**.
>
> > **Q2.** How computationally expensive is D-TSN? How does it scale w.r.t. the action space size and the search tree depth?
>
> We report the average time taken per training step and the total training time for a sample Procgen environment on a RTX 2080Ti GPU in the following table:
> | Solver               | Time Taken Per Training Step | Total Training Time |
> |----------------------|------------------------------|---------------------|
> | Model-free Q-network | 43ms                         | 4h0m                |
> | TreeQN (depth=2)     | 357ms                        | 23h45m              |
> | D-TSN (n_itr=5)      | 153ms                        | 9h30m               |
> | D-TSN (n_itr=10)     | 294ms                        | 17h30m              |
>
> The computational cost of D-TSN grows linearly as the size of the action space increases. Since D-TSN uses best-first search, it **does not have a fixed search tree depth**. We quantify the complexity of the tree by the number of search iterations performed. Given this search budget, D-TSN may learn to search deeper or wider depending on the specific problem. Lastly, the computational cost of D-TSN grows linearly as the number of search iterations increases.
>
> We sincerely hope that our clarifications above have increased your confidence in our work. We will be happy to clarify further if needed. We thank you again for sharing your valuable feedback on our work.
>
> [1] Antonoglou, Ioannis et al. “Planning in Stochastic Environments with a Learned Model.” International Conference on Learning Representations (2022).

---

> > ### Comment · Reviewer_tL81 · 2024-11-25
> >
> > Thank you for your thorough answers and precise explanation.
> >
> > > Since D-TSN uses best-first search, it does not have a fixed search tree depth.
> >
> > Makes sense. I think a better question would be about the depth of the solution, which I now understand thanks to your explanation, the provided table, and you other clarification that a fixed number of nodes/iterations is used.
> >
> > The authors' responses have reassured me in my decision, and I will proceed with the same score.
> >
> > Good luck with everything ahead.

---

### Meta-Review · Area_Chair_r2ws · 2024-12-20

**Metareview:**

This paper introduces a new differentiable neural tree search architecture that learns directly from data trajectories, effectively embedding a search-like inductive bias into the neural network weights. The method addresses a critical limitation of the existing TreeQN method, which allows it to work with stochastically sampled tree structures instead of relying on a fixed tree. The authors demonstrate the approach's effectiveness through empirical comparisons against model-free and model-based methods, as well as TreeQN, showcasing improved performance across several reinforcement learning environments.

This paper received universal positive scores after the rebuttal period. All reviewers agreed that the approach is novel, and that the empirical results were very compelling across the wide range of environments tested. Concerns about the framing and computational complexity / speed were largely addressed during the rebuttal period. While the presented method is not directly applicable to stochastic environments, the authors presented an example of how the approach could be adapted to this setting during the rebuttal.

Given the compelling originality of the approach, and its clear empirical success, I recommend acceptance. Since the approach will likely be of broad interest to anyone using search (which is likely a large audience at ICLR), I am recommending a spotlight presentation.

**Additional Comments On Reviewer Discussion:**

The discussion was very productive between authors and reviewers. Several reviewers made suggestions to improve the manuscript which were incorporated into the revision. The final paper after the revision period is clear, well-written, and better contextualizes the work and the results, as also noted by the reviewers.

---

### Decision · Program_Chairs · 2025-01-22

Accept (Oral)